# Omni-View: Unlocking How Generation Facilitates Understanding in Unified 3D Model based on Multiview images

**JiaKui Hu**[1,2*] **Shanshan Zhao**[2†] **Qing-Guo Chen**[2] **Xuerui Qiu**[3,4] **Jialun Liu**[5]
**Zhao Xu**[2] **Weihua Luo**[2] **Kaifu Zhang**[2] **Yanye Lu**[1†]
[1]Institute of Medical Technology, Peking University
[2]Alibaba International Digital Commerce Group [3]CASIA [4]ZGCA [5]TeleAI

## Abstract

This paper presents Omni-View, which extends the unified multimodal understanding and generation to 3D scenes based on multiview images, exploring the principle that "generation facilitates understanding". Consisting of understanding model, texture module, and geometry module, Omni-View jointly models scene understanding, novel view synthesis, and geometry estimation, enabling synergistic interaction between 3D scene understanding and generation tasks. By design, it leverages the spatiotemporal modeling capabilities of its texture module responsible for appearance synthesis, alongside the explicit geometric constraints provided by its dedicated geometry module, thereby enriching the model's holistic understanding of 3D scenes. Trained with a two-stage strategy, Omni-View achieves a state-of-the-art score of 55.4 on the VSI-Bench benchmark, outperforming existing specialized 3D understanding models, while simultaneously delivering strong performance in both novel view synthesis and 3D scene generation. The code and pretrained models are open-sourced at https://github.com/AIDC-AI/Omni-View.

## 1 Introduction

Recently, unified multimodal models (UMMs) (Team, 2024a; Xie et al., a; Deng et al., 2025) have emerged as a pivotal area of research. The primary goal is to empower multimodal large language models (MLLMs) to both understand and generate visual signals present in our world, laying the foundation for artificial general intelligence.

Numerous methods (Team, 2024a; Wu et al., 2024a; b; Xie et al., a; Chen et al., 2025; Deng et al., 2025; Li et al., 2025) have achieved coexistence between 2D image understanding and generation. Among them, some studies (Wu et al., 2024b; Tong et al., 2024) aim to improve model performance by exploring the synergies arising from the interaction between understanding and generation. For example, Metamorph (Tong et al., 2024) has extensively validated the role of understanding in improving generation performance. However, the potential and effectiveness of generation to improve understanding capabilities within unified models remain underexplored.

Due to the intrinsic geometric and spatiotemporal nature of 3D scenes and their multiview images, generative tasks such as geometry estimation and novel view synthesis are particularly well-suited for facilitating understanding in the 3D domain. This is attributable to the fact that 3D understanding tasks (Yang et al., 2025; Zhang et al., a) inherently necessitate robust geometric and spatiotemporal modeling capabilities, which can be effectively acquired through these generative tasks. Moreover, biomedical evidence (Maus et al., 2013; Nortmann et al., 2015) suggests that human understanding of 3D environments is governed by the capacity to generate and imagine future sensory and geometric data (Keller et al., 2012; Leinweber et al., 2017) in the observed scene. These findings provide us with a guidance: the "generation facilitates understanding" paradigm presents a promising approach to building a unified model for 3D scene understanding and generation.

---

*This work is done when he interns at Alibaba.
†Corresponding authors, yanye.lu@pku.edu.cn, sshan.zhao00@gmail.com

Inspired by the above analysis, this paper aims to fully unlock and maximize the benefits of generation to understanding, thereby constructing a unified model for 3D scene understanding and generation, called **Omni-View**. We emphasize that geometry estimation and novel view synthesis can leverage their inherent geometric measurement and spatiotemporal modeling capabilities to improve 3D scene understanding, localization, and spatial reasoning. Specifically, we achieve this through two key aspects, *architectural design* and *training strategy*.

The *architectural design* aims at unifying 3D scene understanding and generation, leveraging geometry estimation and novel view synthesis to advance 3D scene understanding. Our Omni-View is built on Bagel (Deng et al., 2025), a strong unified framework in which the interaction between understanding and generation is facilitated by its shared multimodal self-attention. However, Bagel's generative capacity is limited to RGB images, thus only capturing texture information, whereas 3D scene generation necessitates inclusion of both texture and geometric structure. In accordance with the dual generative objectives, the generation model in Omni-View is split into two distinct modules: a texture module and a geometry module. The texture module receives reference images, a list of targeted camera poses, and prompt tokens encoded by the understanding model to generate novel views of the scene. Meanwhile, the geometry module employs the hidden features from the understanding model and the latent output of the texture module to infer geometric information of novel views, such as depth maps and camera poses. This dual-pathway architecture empowers the model to develop both geometric and spatiotemporal modeling capabilities, which are essential for 3D scene understanding tasks.

The principal goal of *training strategy* is to comprehensively improve the performance of the model. A two-stage training strategy is employed. The first stage aims to augment the benefits of generation for understanding 3D scenes, as introduced by the proposed architecture. The subsequent stage is intended to refine the generation performance. In stage 1, the understanding model, geometry module, and texture module are trained simultaneously. Geometry estimation assists the model in comprehending the relative positional relationships among objects, thus directly enhancing the model's capability to evaluate the relative distances and directions of objects in 3D scenes. The autoregressive generation forces the understanding model to discern the spatiotemporal relations between the generated novel views, thus improving its understanding capabilities. As iterations progress in stage 1, the number of reference images gradually decreases. This progressive shift from dense to sparse views supports a curriculum-like, easy-to-difficult training approach, ultimately enhancing the performance of the understanding model. In stage 2, the understanding model is frozen. The generation model is finetuned via RGB-Depth-Pose joint generation, thereby enhancing its capabilities in 3D scene generation.

We evaluate Omni-View on scene understanding, spatial reasoning, and novel view synthesis tasks. The model achieves an impressive score of 55.4 on the VSI-Bench, exceeding current MLLMs designed for visual reasoning. It manifests particularly notable improvements in subtasks such as Relative Distance and Appearance Order, which require spatiotemporal modeling and the estimation of geometry acquired through generation tasks. It also exhibits superior performance compared to existing 3D understanding MLLMs in the area of 3D question answering (Ma et al., 2023; Azuma et al., 2022). Furthermore, it effectively narrows the performance gap between unified models and specialized models focused on 3D visual localization (Chen et al., 2020). Furthermore, we attain robust results in the domain of novel view synthesis and scene generation, with particular emphasis on enhanced perceptual quality.

## 2 RELATED WORK

**Scene understanding.** Recent advances in understanding 3D scenes have been greatly provided by incorporating 3D or video input and 3D reconstruction prior. LLaVA-3D (Zhu et al., 2025) and GPT4Scene (Qi et al., 2025) function within the voxel space and BEV. Video3DLLM (Zheng et al., 2025) improves localization ability by encoding 3D coordinates as position embeddings, while Ross3D (Wang et al., 2025a) improves 3D understanding through visual-centric reconstruction. However, the dependency on 3D input of these methods poses practical application challenges. To alleviate this issue, VG-LLM (Zheng et al.) and SpatialMLLM (Wu et al., a) use the features of VGGT (Wang et al., 2025b) as input, embedding the 3D piror in the model.

**Scene generation.** A vital element of scene generation methods is the presence of an efficient proxy to represent the 3D scene, with panoramic images (Team et al., 2025), point clouds (Yu et al., 2025b), and Gaussian Splatting (Yu et al., 2025a) among the viable options. In consideration of the advances in image and video diffusion models (Rombach et al., 2022; Wan et al., 2025), contemporary 3D scene generation strategies reconstruct the scene's geometry from a single view, employing conditioned video diffusion models to render the scene's texture. ViewCrafter (Yu et al., 2025b) brings explicit 3D information (point cloud) to generation models through iterative reconstruction. WonderJourney (Yu et al., 2024) generates a comprehensive set of view sequences. Voyager (Huang et al., 2025) reconstructs the scene from a single view and uses it as conditions for inpainting.

**Unified understanding and generation.** In the domain of 3D scenes, a unified model for understanding and generation applicable in general scenarios remains absent. Hermes (Zhou et al., 2025) uses BEVs to design a unified understanding and generation model for autonomous driving. In contrast, significant progress has been made in 2D vision (Zhang et al., 2025). These methods show primarily variations in architectures and training strategies. Chameleon (Team, 2024a) utilizes VQ-VAE for image tokenization, thereby improving generation competencies. However, its understanding capabilities are inferior to those of Janus (Wu et al., 2024a), which employs SigLIP (Zhai et al., 2023) as visual understanding encoder. VILA-U (Wu et al., b) integrates understanding and generation within the image encoder, bypassing multitask gradient conflicts during MLLM training. BAGEL (Deng et al., 2025) implements task-based hard routing for MLLM, which also avoids gradient conflicts. Harmon (Wu et al., 2025) takes advantage of MAE's reconstruction ability and downstream understanding enhancement. BLIP3o (Chen et al., 2025) introduces "understand first, then generation" training paradigm, thus achieving performance gains. Building on the progress in the 2D domain, this paper investigates a unified model in 3D scenes and explores how generation aids the understanding scheme within the framework.

## 3 METHOD

### 3.1 ARCHTITECTURES

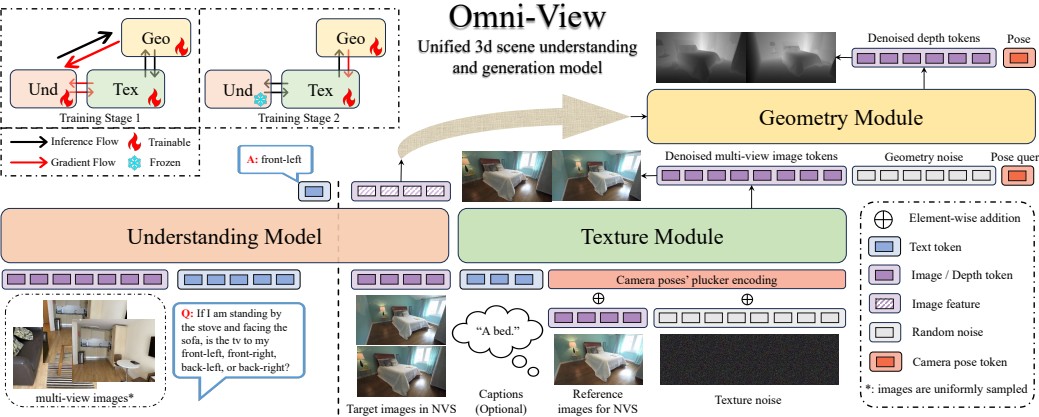

Figure 1: **Architecture of Omni-View**. Building upon Bagel (Deng et al., 2025), Omni-View consists of an understanding model and a generation model. The generation model is further composed of two specialized modules: one for texture and one for geometry. Trained via a two-stage process, Omni-View shows high effectiveness in scene understanding and novel view synthesis. Crucially, it unlocks the benefits of its generative capabilities to enhance the model's understanding performance.

The architecture of Omni-View, as illustrated in Figure 1, is divided into two distinct components. The left component, termed the understanding model, is tasked with understanding the multiview images of 3D scenes and producing text feedback. The right component is comprised of two integral parts: the texture module and the geometry module, which facilitate the synthesis of the RGB image and the associated geometric data.

**Understanding Model.** The understanding model is required to perform 3D scene or spatial comprehension tasks, including question answering, localization, and reasoning. This necessitates the

model to extract semantic information from spatial-temporal data while measuring the geometric characteristics of the 3D scene. Specifically, given $n$ multiview images and a question $T_{ques}$, the understanding model produces the textual answer $T_{ans}$ through the next token prediction. In this process, the images are encoded by SigLIP (Zhai et al., 2023) and the text is tokenized by a frozen vocabulary $\tau(\cdot)$. Subsequently, tokens of multiview images and question $\tau(T_{ques})$ will be sequentially concatenated and fed into the understanding model. Moreover, the understanding model can also produce intermediate features, which will be used in the geometry module.

**Texture Module.** The texture module in the generation model is tasked with novel view synthesis using flow matching (Lipman et al.). This module processes a textual description $T_{des}$ and some reference images $I_{ref}$ of the current scene, together with a specified camera pose, to produce consistent novel views of this scene. Within this module, the reference images $I_{ref}$ are encoded using the FLUX-VAE encoder (Labs, 2024) $\varepsilon(\cdot)$, and the vocabulary used to tokenize $T_{des}$ aligns with that used by the understandimg model. This process incorporates the camera pose control as delineated in MV-AR (Hu et al., 2025), embedding the Plucker-Ray encoding $r_{i,j} = (o \times d, d)$ of the camera pose as the absolute position encoding in the model, where $o$ and $d$ denote the origin and direction of the ray, $(i, j)$ represents the pixel coordinate. This camera pose embedding exhibits adaptability to various image resolutions and demonstrates significant flexibility. The above process can be described as:

$$F_{tex} = TextureModule\left([LM\text{-}Head(\tau(T_{des})); [\varepsilon(I_{ref}); N_{tex}] + r]\right), \tag{1}$$

where $F_{tex}$ is the predicted image noise of the texture module, $N_{tex}$ is the random input noise, and LM Head is the text processing module of the understanding model.

**Geometry Module.** The geometry module in the generation model constructs the geometric aspects of the generated images in the texture module. It synthesizes depth maps through flow matching and employs a learnable query strategy to accurately estimate camera poses. It receives the latent of novel view images $F_{tex}$ from the texture module as input, which is concatenated with random depth noise $N_{dep}$ and a learnable camera pose query $q_{cam}$ along the frame dimension. After processing, $N_{dep}$ will be denoised as depth maps' latent, while $q_{cam}$ will be decoded to reveal the intrinsic and extrinsics of the camera. The features of novel view images $F_{und}$, output from the understanding model's block, is also integrated into the geometry module through cross-attention. The process in the geometry module can be described as follows.

$$[F_{dep}; \hat{g}] = GeometryModule\left([F_{tex}; N_{dep}; q_{cam}], F_{und}\right), \tag{2}$$

where $F_{dep}$ is the predicted depth noise of the geometry module. $\hat{g}$ is the predicted intrinsics and extrinsics of the camera, which is decoded by VGGT's camera decoder (Wang et al., 2025b).

The geometry module has independent parameters and maintains architectural connections with both the texture module and the understanding model. Unlike the connection method in BAGEL, the geometry module uniformly extracts features from the understanding model at the layer dimension as conditions. It utilizes cross-attention for fusion and ensures that the gradients of geometry estimation can be backpropagated to the understanding model. Currently, the geometry module's input only relies on the last-layer output latent of the texture module. This approach guarantees that it acquires information closest to the image modality, thereby providing finely aligned features for the estimation of depth map and camera pose.

## 3.2 TRAINING RECIPE

The Omni-View training recipe has two separate stages, as shown in the upper left of Figure 1.

**Stage 1: unify 3D understanding and generation.** In stage 1, we train the understanding model, the texture module, and the geometry module simultaneously. It can leverage the fine-grained geometry estimation capability in the geometry module and combines it with the spatial-temporal modeling ability within the texture module, thus improving the 3D understanding performance.

**Understanding model.** For the understanding model, we use the next token prediction to predict the distribution of the answer text given the distribution of the multiview images and query text. The loss function of the understanding model can be expressed as follows.

$$L_{und} = -\sum_{i=1}^{T} log P_\theta(y_i|y_{<i}),$$ (3)

where $y$ is the multimodal sequence that contains tokenized multiview images, query text, and answer text.

**Texture module.** The loss function for the texture module is defined as the Mean Squared Error (MSE) loss between each predicted texture noise $F_{tex}$ generated by the texture module and the provided texture noise $N_{tex}$.

$$L_{tex} = ||F_{tex} - N_{tex}||_2.$$ (4)

In contrast to the majority of novel view synthesis methods, the texture model employs a autoregressive generation framework. Specifically, during the generation of the $n$-th frame, the model is exposed to the visual data of the preceding $n-1$ frames, while excluding any subsequent frames. This autoregressive methodology enables the model to fully grasp the concept of temporal sequences, thereby enhancing its spatiotemporal modeling proficiency and effectively improving scene understanding. To optimize the 3D consistency of the sampled images, diffusion forcing (Chen et al., 2024a) is employed when training the texture module.

To acquire this complex spatiotemporal generation capability incrementally, we gradually adjust the reference images. As iterations progress, the reference images are systematically reduced in a stepwise manner, transitioning from encompassing all input images, excluding the first image, to including only the first image. This implies that for the model, the reference confidence transitions from dense to sparse. We designate this progressive training approach as dense-to-sparse (D2S). This strategy has been shown to be highly effective in facilitating improved understanding.

**Geometry module.** In stage 1, the geometry module is tasked with estimating the depth and pose of the camera from both the provided images and those synthesized by the texture module. The loss function for the geometry module comprises a sum of the depth estimation loss and the camera pose estimation loss. In terms of depth estimation, we apply the MSE loss to the comparison between the depth noise $F_{dep}$ predicted by the geometry module and the given depth noise $N_{dep}$. The estimated intrinsics and extrinsics of the camera $\hat{g}$ are optimized directly through the Huber loss.

$$L_{geo} = ||F_{dep} - N_{dep}||_2 + ||\hat{g} - g_{gt}||_\epsilon,$$ (5)

where $g_{gt}$ is the ground truth of the camera pose and $|| \cdot ||_\epsilon$ denotes the Huber loss.

Ultimately, a weighted summation of the aforementioned losses is conducted to derive the loss function in stage 1 $L_{s1}$.

$$L_{s1} = \lambda_{und}L_{und} + \lambda_{tex}L_{tex} + \lambda_{geo}L_{geo},$$ (6)

where $\lambda_{und}, \lambda_{tex}$, and $\lambda_{geo}$ represent the weighting coefficients and their default values are 1, 1, and 0.1 respectively.

**Stage 2: advance generation.** In stage 2, the texture module and the geometry modules are trained. The RGB-Depth-Pose (RGBDP) joint learning methodology is used for training, capitalizing on the geometry prior obtained from depth-pose estimation to enhance the ability to generate consistent appearances for novel views.

In the texture module, the reference single-view image is used along with its depth map to reconstruct the initial point cloud of the scene. The images rendered from different views, projected through this point cloud, serve as conditions following Voyager (Huang et al., 2025). For the geometry module, it generates the depth map and camera pose from images synthesized by the texture module. Concurrently, it no longer relies on the features of the understanding model as conditions for cross-attention.

## 4 EXPERIENCE

### 4.1 EXPERIMENTAL SETUP

In our experiments, the understanding model and the texture module in the generation model are initialized using the pre-trained BAGEL-7B (Deng et al., 2025). The geometry module in the generation model is configured to have the same dimensions as the texture module, with a depth of four layers. For 3D scene understanding, a filtered dataset comprising 780k valid items, sourced from SQA3D (Ma et al., 2023), ScanQA (Azuma et al., 2022), 3DOD (Zheng et al.), ScanRefer (Chen et al., 2020), VLM-3R (Fan et al., 2025), SPAR (Zhang et al., a) (234k subset) and llava-hound4 (Zhang et al., 2024b) (64k subset), is meticulously curated for training. Regarding novel view synthesis, 61k video clips are carefully selected from re10k (Zhou et al., 2018). The corresponding depth maps are synthesized using the Voyager data pipeline (Huang et al., 2025) and captions are synthesized using the QwenVLMax (Bai et al., 2025). During training, we do not use images from the scene understanding task to train the generation model to fully demonstrate that the understanding performance improvement brought by Omni-View does not come from memorizing the data in understanding tasks.

Model training was performed using the AdamW optimizer, characterized by $\beta_1 = 0.9, \beta_2 = 0.95$, the peak learning rate of $1 \times 10^{-5}$. The warm-up phase that constitutes 5% of the whole training iterations. The training process is completed after one epoch of the understanding dataset. For the understanding model, we do not rely on any 3D scene input to support both 3D scene understanding and spatial reasoning tasks. In main comparisons, we use the same checkpoint for testing all tasks.

### 4.2 MAIN COMPARISONS

#### 4.2.1 3D SCENE UNDERSTANDING

**Benchmarks.** We evaluate the 3D understanding performance of the model on question answering (Ma et al., 2023; Azuma et al., 2022) and localization (Chen et al., 2020; Zheng et al.). During inference, we set the frame numbers as 32 following Video3DLLM (Zheng et al., 2025).

**Comparison baselines.** We compare Omni-View with models specifically designed for 2D or 3D visual understanding tasks (Team, 2024b; Wang et al., 2024a; Zhang et al., 2024b; Huang et al., 2023; Zhang et al., 2024a; Chen et al., 2024b; Zhu et al., 2025; Zheng et al., 2025; Qi et al., 2025; Wang et al., 2025a), as well as with some unified models applicable to video modalities (Deng et al., 2025). Unified models are evaluated after fine-tuning with the same data. Within the results table, we differentiate between models that incorporate explicit 3D input and those that do not. Although the incorporation of explicit 3D input improves model performance, it concurrently restricts applicability (Zheng et al., 2025).

**Metrics.** For SQA3D, we use the EM metric to evaluate the accuracy, which stands for top-1 exact match. EM-R means the refined EM following LEO (Huang et al., 2023). For ScanQA, we use CIDEr (C), BLEU-4 (B-4), METEOR (M), ROUGE (R), and EM for more complete validation. For 3DOD, we use the average F1 score (F1) as a metric to assess the correspondence between the predicted and actual coordinates. For ScanRefer, we calculate the percentage of samples for which the Intersection over Union (IoU) exceeds thresholds of 0.25 and 0.5, respectively, between the predicted and true coordinates. The higher the above metrics, the better.

**Results.** As shown in Table 1, our analysis leads to four conclusions. (1) Our Omni-View exceeds all current MLLM methods that do not depend on 3D scene input. Within the SQA3D test set, Omni-View achieves an enhancement of 3.3 over SpatialMLLM in EM, while in the ScanQA validation set, it surpasses SpatialMLLM by an increment of 11.2 in CIDEr. For the object detection task, our unified model performs comparably with the 3D understanding model VG-LLM. (2) This performance improvement is mainly attributed to the architectural design and training scheme that we proposed. Compared with the fine-tuned BAGEL, our method still demonstrates substantial improvements. For example, on the SQA3D test set, Omni-View enhances EM by 2 points compared to the fine-tuned BAGEL (Deng et al., 2025); on the ScanQA validation set, Omni-View improves 7.5 in CIDEr. (3) The efficacy of our approach in the question-answering task is equivalent to advanced MLLM methods that require 3D scene input. In particular, in the ScanQA validation set,

| Methods | 3D Input | SQA3D$_{test}$ | | ScanQA$_{val}$ | | | | | 3DOD | ScanRefer | |
|---|---|---|---|---|---|---|---|---|---|---|---|
| | | EM | EM-R | C | B-4 | M | R | EM | F1 | Acc@0.25 | Acc@0.5 |
| *Task-specific Models* | | | | | | | | | | | |
| LEO | ✔ | 50.0 | 52.4 | 80.0 | 11.5 | 16.2 | 39.3 | 21.5 | – | – | – |
| ChatScene | ✔ | 54.6 | 57.5 | 87.7 | 14.3 | 18.0 | 41.6 | 21.6 | – | 55.5 | 50.2 |
| Grounded 3D-LLM | ✔ | – | – | – | – | – | – | – | – | 47.9 | 44.1 |
| Video-3D-LLM | ✔ | 58.6 | – | 102.1 | 16.4 | 20.0 | 49.3 | 30.1 | – | 58.1 | 51.7 |
| GPT4Scene-HDM | ✔ | 59.4 | 62.4 | 96.3 | 15.5 | 18.9 | 46.5 | – | – | 62.6 | 57.0 |
| Ross3D | ✔ | 63.0 | 65.7 | 107.0 | 17.9 | 20.9 | 50.7 | 30.8 | – | 61.1 | 54.4 |
| LLaVA-3D | ✔ | 60.1 | – | 103.1 | 16.4 | 20.8 | 49.6 | 30.6 | – | 50.1 | 42.7 |
| InternVL2-8B | ✘ | 33.0 | 45.3 | 62.5 | 3.3 | 14.5 | 34.3 | – | – | – | – |
| Qwen2-VL-7B | ✘ | 48.5 | – | 53.9 | 3.0 | 11.4 | 29.3 | – | – | – | – |
| LLaVA-Video-7B | ✘ | 48.5 | – | 88.7 | 3.1 | 17.7 | 44.6 | – | – | – | – |
| SPAR-7B | ✘ | 58.1 | – | 90.7 | 15.3 | – | – | – | – | 48.8 (31.9) | 43.1 (12.4) |
| VG-LLM-4B | ✘ | – | – | – | – | – | – | – | 47.0 | 53.5 (36.4) | 47.5 (11.8) |
| SpatialMLLM-4B | ✘ | 55.9 | 58.7 | 91.8 | 14.8 | 18.4 | 45.0 | – | – | – | – |
| *Unified Models* | | | | | | | | | | | |
| BAGEL-7B-FT | ✘ | 57.2 | 59.7 | 95.5 | 14.7 | 18.7 | 46.3 | 27.0 | 41.3 | 46.9 (28.0) | 41.6 (7.7) |
| Omni-View-7B | ✘ | **59.2** | **61.9** | 103.0 | 16.2 | 20.1 | **49.0** | **29.5** | 46.4 | 50.8 (32.5) | 45.0 (9.9) |

Table 1: **Evaluation of 3D scene understanding**. "–" indicates the number is not available for us. **Bold** and underline denote the best and second-best models without 3D scene input, respectively. For ScanRefer, the content in "()" indicates results without proposal refinement (Zhang et al., a).

Omni-View achieved a performance comparable to Video3DLLM (Zheng et al., 2025) and LLaVA-3D (Zhu et al., 2025). (4) However, there remains a notable disparity between methodologies that do not require 3D scene input and those that do, particularly in 3D grounding tasks. The experimental results of VG-LLM (Zheng et al.) also substantiate this observation.

### 4.2.2 3D SPATIAL REASONING

**Benchmarks.** We evaluate the 3D spatial reasoning ability of the model in VSI-Bench (Yang et al., 2025). During inference, we follow the VSI-Bench to set frame numbers ranging from 8 to 32 and frame resolution to 640p.

**Comparison baselines.** We compare our Omni-View with models specifically designed for 2D or 3D visual understanding tasks (Chen et al., b; Zhang et al., b; Bai et al., 2025; Ray et al.; Zhang et al., a; Zheng et al.; Wu et al., a), as well as with some unified models applicable to video modalities (OpenAI, 2024; Team et al., 2024; Deng et al., 2025; Xie et al., b). Unified models are evaluated after fine-tuning with the same data we used.

| Methods | Numerical Qusetion | | | | Multiple-Choice Question | | | | Avg. |
|---|---|---|---|---|---|---|---|---|---|
| | Obj. Cnt. | Abs. Dist. | Obj. Size | Room Size | Rel. Dist. | Rel. Dir. | Route Plan | Appr. Order | |
| *Task-spefic Models* | | | | | | | | | |
| LongVILA-8B | 29.1 | 9.1 | 16.7 | 0.0 | 29.6 | 30.7 | 32.5 | 25.5 | 21.6 |
| LongVA-7B | 38.0 | 16.6 | 38.9 | 22.2 | 33.1 | 43.3 | 25.4 | 15.7 | 29.2 |
| LLaVA-OneVision-72B | 43.5 | 23.9 | 57.6 | 37.5 | 42.5 | 39.9 | 32.5 | 44.6 | 40.2 |
| LLaVA-Video-72B | 48.9 | 22.8 | 57.4 | 35.3 | 42.4 | 36.7 | 35.0 | 48.6 | 40.9 |
| Qwen2.5VL-7B | 40.9 | 14.8 | 43.4 | 10.7 | 38.6 | 38.5 | 33.0 | 29.8 | 33.0 |
| Qwen2.5VL-72B | 25.1 | 29.3 | 54.5 | 38.8 | 38.2 | 37.0 | 34.0 | 28.9 | 37.0 |
| SAT-LLaVA-Video-7B | – | – | – | 47.3 | 41.1 | 37.1 | **36.1** | 40.4 | – |
| SPAR-8B | – | – | – | – | – | - | - | – | 41.1 |
| VG-LLM-4B | 66.4 | 36.6 | 55.2 | **56.3** | 40.8 | 43.4 | 30.4 | 39.5 | 46.1 |
| Spatial-MLLM-4B | 65.3 | 34.8 | 63.1 | 45.1 | 41.3 | 46.2 | 33.5 | 46.3 | 48.4 |
| *Unified Models* | | | | | | | | | |
| GPT-4o (API) | 46.2 | 5.3 | 43.8 | 38.2 | 37.0 | 41.3 | 31.5 | 28.5 | 34.0 |
| Gemini-1.5 Pro (API) | 56.2 | 30.9 | 64.1 | 43.6 | 51.3 | 46.3 | 36.0 | 34.6 | 45.4 |
| BAGEL-7B-FT | 62.8 | 36.3 | 56.4 | 49.7 | 46.1 | 49.4 | 26.8 | 43.1 | 46.3 |
| Omni-View-7B | **70.3** | **46.4** | 68.6 | 54.7 | **65.9** | **54.4** | 33.5 | **49.0** | **55.4** |

Table 2: **Evaluation of spatial reasoning on VSI-Bench**. "–" indicates the number is not available for us. **Bold** and underline denote the best and second-best models, respectively.

**Results.** The results on spatial reasoning tasks more fully demonstrate Omni-View's improvement over previous methods in analyzing the relative or absolute position and orientation of spatial ob-

jects. With an average score of 55.4, our Omni-View ranks first among existing spatial reasoning MLLMs. Compared to existing Spatial-MLLM (Wu et al., a) and VG-LLM (Zheng et al.), Omni-View improves Spatial-MLLM by 11.6, 9.6, and 24.6 in Absolute Distance (Abs. Dist.), Room Size, and Relative Distance (Rel. Dist.), respectively. Omni-View improves VG-LLM by 9.8, 13.4, 25.1, 11.0, and 9.5 in Abs. Dist., Object Size (Obj. Size), Rel. Dist., Relative Direction (Rel. Dir.), and Appearance Order (Appr. Order), respectively. These tasks necessitate the model to thoroughly predict the spatiotemporal state and measure the geometric properties of the scene it observes (Yang et al., 2025). This demonstrates that our proposed method substantially enhances the model's pertinent capabilities.

### 4.2.3 3D SCENE GENERATION

**Benchmarks.** The model's generation capacity is validated under two tasks: novel view synthesis (NVS) from a single view and scene generation. NVS from a single view necessitates that the model generate the subsequent 25 frames from the first image. 3D scene generation evaluates scenes that are reconstructed from the videos generated to 3DGS. The test scenes are randomly selected from the Re10k test set following Chen et al. (a).

**Comparison baselines.** We compare with scene generation methods (Chung et al., 2025; Wang et al., 2024b; Ma et al., 2025; Yu et al., 2025b; Chen et al., a; Huang et al., 2025) and a unified model (Deng et al., 2025). To evaluate the performance of scene generation, we use Dust3R to reconstruct the generated videos, ensuring fairness in the comparison, according to Chen et al. (a). The evaluation is only conducted on the first 25 frames generated by the model following Yu et al. (2025b); Zhai et al. (2025).

**NVS from single view and scene generation.** Omni-View achieved the highest PSNR and SSIM, and lowest LPIPS score, indicating that its image quality could surpass that of other methods. However, in terms of pixel-level fidelity, Omni-View shows only slight improvements over popular scene generation models. This discrepancy may be attributed to Omni-View's challenges in being precisely controlled via the camera pose. Visualization results are presented in the Appendix A.4.

| Methods | NVS from Single View | | | Scene Generation | | |
|---|---|---|---|---|---|---|
| | PSNR ↑ | SSIM ↑ | LPIPS ↓ | PSNR ↑ | SSIM ↑ | LPIPS ↓ |
| *Task-specific Models* | | | | | | |
| LucidDreamer | 22.27 | 0.766 | 0.204 | 21.98 | 0.698 | 0.290 |
| MotionCtrl | 15.86 | 0.520 | 0.431 | 15.33 | 0.479 | 0.590 |
| See3D | 22.37 | 0.781 | 0.199 | 21.60 | 0.744 | 0.238 |
| ViewCrafter | 22.60 | 0.754 | 0.195 | 22.25 | 0.709 | 0.204 |
| FlexWorld-5B | 23.05 | 0.788 | 0.182 | 22.86 | 0.756 | 0.198 |
| Voyager-13B | 23.12 | 0.793 | 0.175 | 22.93 | 0.768 | 0.194 |
| *Unified Models* | | | | | | |
| BAGEL-7B-FT | 21.76 | 0.703 | 0.288 | 21.04 | 0.599 | 0.403 |
| Omni-View-7B | **23.22** | **0.817** | **0.114** | **23.12** | **0.801** | **0.146** |

Table 3: **Evaluation of novel view synthesis from single view and scene generation on Re10k.** **Bold** and underline denote the best and second-best models, respectively.

### 4.3 ABLATION STUDIES

We perform ablation studies on the proposed architecture and training strategy to ascertain their efficacy. The data used and the hyperparameters applied during the ablation studies remain consistent.

**Effect of two modules in the generation model.** Both the texture module and the geometry module can improve understanding performance, but they focus on different aspects. The results presented in Table 4 demonstrate that the integration of the texture module facilitates notable advancements in tasks dependent on spatiotemporal modeling, exemplified by an increase of 4.1 points in Appr. Order. Conversely, the introduction of the geometry module markedly enhances performance in tasks contingent upon relative position information, notably in Rel. Dist. However, because of the lacking absolute metric in the synthesized depth maps, improvements in tasks pertaining to absolute metric comprehension, such as Abs. Dist., are constrained. Incorporating the task of accurately predicting camera pose can mitigate the reduction resulting from the imprecise depth prediction. Furthermore,

our results corroborate that the segregation of the texture and geometry modules results in superior understanding performance compared to employing a unified architecture that concurrently learns both texture and geometry.

| Texture | Geometry | | SQA3D | ScanQA | | ScanRefer | VSI-Bench (subset) | | | | |
|---|---|---|---|---|---|---|---|---|---|---|---|
| | depth | camera | EM | C | B-4 | Acc@0.25 | Obj. Cnt. | Abs. Dist. | Obj. Size | Rel. Dist. | Appr. Order |
| ✗ | ✗ | ✗ | 57.2 | 95.5 | 14.7 | 46.9 (28.0) | 62.8 | 36.3 | 56.4 | 46.1 | 43.1 |
| ✔ | ✗ | ✗ | 58.3 | 97.4 | 14.7 | 48.8 (31.5) | 67.7 | 44.6 | 59.0 | 53.2 | 47.2 |
| ✔ | ✔ | ✗ | 58.9 | 100.5 | 16.0 | 48.2 (31.2) | 69.0 | 44.9 | 69.7 | 63.0 | 47.9 |
| ✔ | ✔ | ✔ | 58.7 | 99.2 | 15.0 | 49.0 (31.6) | 69.5 | 45.9 | 67.8 | 63.8 | 48.2 |
| ✔ | ✔ | ✔ | 59.2 | 103.0 | 16.2 | 50.8 (32.5) | 70.3 | 46.4 | 68.6 | 65.9 | 49.0 |

Table 4: **Ablation on modules in the generation model.** The gray row denotes that, in this experiment, both the texture module and the geometry module use the same architecture and parameters.

**Effect of the autoregressive generation.** Autoregressive generation significantly improves tasks requiring spatiotemporal modeling, like Appr. Order. Table 5 compares the understanding performance with and without autoregressive generation. Although bidirectional generation offers some enhancement, autoregressive generation provides greater improvements. Specifically, it boosts performance by 5.8 and 4.4 points in Abs. Dist. and Appr. Order tasks, respectively, compared to when not used. This demonstrates that autoregressive generation strengthens spatiotemporal modeling and enhances related scene understanding tasks.

| Autoregressive | SQA3D | ScanQA | | ScanRefer | VSI-Bench (subset) | | | | |
|---|---|---|---|---|---|---|---|---|---|
| | EM | B-4 | EM | Acc@0.25 | Obj. Cnt. | Abs. Dist. | Obj. Size | Rel. Dist. | Appr. Order |
| None | 57.2 | 95.5 | 14.7 | 46.9 (28.0) | 62.8 | 36.3 | 56.4 | 46.1 | 43.1 |
| ✗ | 57.2 | 98.8 | 15.0 | 47.0 (28.0) | 68.3 | 40.6 | 69.0 | 60.6 | 44.6 |
| ✔ | 59.2 | 103.0 | 16.2 | 50.8 (32.5) | 70.3 | 46.4 | 68.6 | 65.9 | 49.0 |

Table 5: Ablation on the autoregressive generation in stage 1. "None" means we only train understanding model in this experiment.

**Effect of the D2S (dense-to-sparse).** The efficacy of D2S training is substantiated during stage 1. The results show that D2S significantly improves the model understanding performance, exceeding the visual reconstruction method advanced in Ross3D (Wang et al., 2025a) for scene understanding tasks necessitating spatiotemporal ordering. In this ablation, the geometry module is kept trainable.

| Condition in S1 | SQA3D | ScanQA | | ScanRefer | VSI-Bench (subset) | | | | |
|---|---|---|---|---|---|---|---|---|---|
| | EM | B-4 | EM | Acc@0.25 | Obj. Cnt. | Abs. Dist. | Obj. Size | Rel. Dist. | Appr. Order |
| None | 57.2 | 95.5 | 14.7 | 46.9 (28.0) | 62.8 | 36.3 | 56.4 | 46.1 | 43.1 |
| randon mask | 58.9 | 98.9 | 15.5 | 49.4 (31.7) | 65.7 | 38.9 | 55.5 | 51.6 | 46.5 |
| dense | 57.3 | 94.7 | 15.0 | 47.2 (28.3) | 67.7 | 42.3 | 60.1 | 65.6 | 46.6 |
| sparse | 57.9 | 97.3 | 15.7 | 50.1 (32.0) | 69.0 | 44.9 | 65.5 | 63.1 | 48.3 |
| dense → sparse (Ours) | 59.2 | 103.0 | 16.2 | 50.8 (32.5) | 70.3 | 46.4 | 68.6 | 65.9 | 49.0 |

Table 6: **Ablation on the D2S mechanism in stage 1 (S1).** "None" means we don't train generation model in this experiment. The "random mask" denotes the visual reconstruction method in Ross3D (Wang et al., 2025a).

**Effect of training stage 2.** The results in Table 7 show that stage 2 can significantly improve the scene generation performance of the model.

| Stage 2 | NVS from single view | | | Scene Generation | | |
|---|---|---|---|---|---|---|
| | PSNR ↑ | SSIM ↑ | LPIPS ↓ | PSNR ↑ | SSIM ↑ | LPIPS ↓ |
| ✗ | 21.90 | 0.705 | 0.265 | 21.44 | 0.683 | 0.307 |
| ✔ | 23.22 | 0.817 | 0.114 | 22.93 | 0.768 | 0.194 |

Table 7: **Ablation of the stage 2.**

## 5    CONCLUSION

We introduce Omni-View, a unified 3D scene understanding and generation model. By decomposing the generation model into distinct texture and geometry components, we establish the feasibility and efficacy of employing generation processes to enhance 3D scene understanding and spatial reasoning. Our method illustrates that a unified understanding and generation model is capable of achieving performance on par with the leading specialized understanding models. We posit that Omni-View can serve as a foundational model across 3D and multiview domains, thereby advancing the development of downstream applications such as spatial intelligence.

**Ethics Statement.** This paper aims to develop unified models to understand and generate 3D scenes. In light of the ongoing advancements in scene generation technology, we emphasize the importance of preventing its misapplication, such as the fabrication of deceptive scenes or the creation of scenes with nefarious intents.

**Reproducibility Statement.** We state that Omni-View is highly reproducible. Implementation details on our main experiences are provided in Section 4.1 and Appendix B. It is anticipated that these descriptions can sufficiently demonstrate the reproducibility of Omni-View. We plan to open-source the code and weight files after the paper passes peer review.

## ACKNOWLEDGEMENT

This work was supported in part by National Key Research and Development Program of China (2025YFA1805700); in part by National Natural Science Foundation of China (82371112, 62501020); in part by the Science Foundation of Peking University Cancer Hospital (JC202505).

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

**Limitations and future work.** Although Omni-View has demonstrated a unified advance in the domains of question answering, spatial reasoning, and novel view synthesis, its grounding capabilities remain to be substantiated. Additionally, its generation model currently lacks the capability for long-range world generation. Future efforts will be concentrated on using reinforcement learning to augment Omni-View's performance in 3D visual grounding and long-range generation. At the same time, since the training data used by its geometry module is a synthetic depth map, its actual geometry prediction ability may not be accurate enough.

# A QUALITATIVE RESULTS

We show some qualitative results for 3D scene understanding and generation tasks in the appendix.

## A.1 3D SCENE UNDERSTANDING

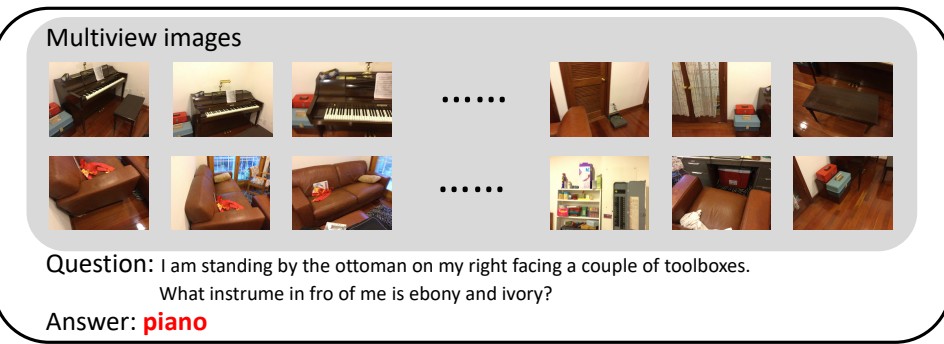

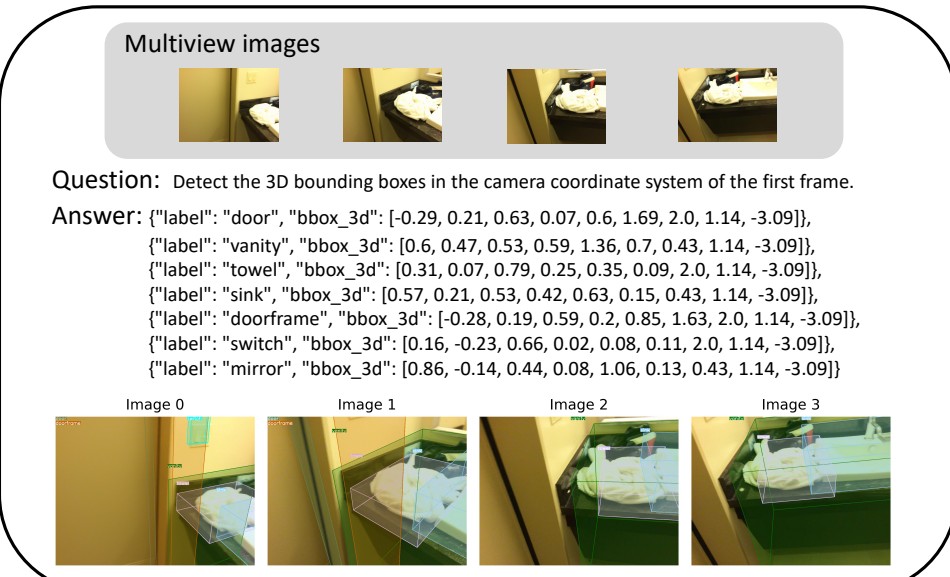

## A.2 SPATIAL REASONING

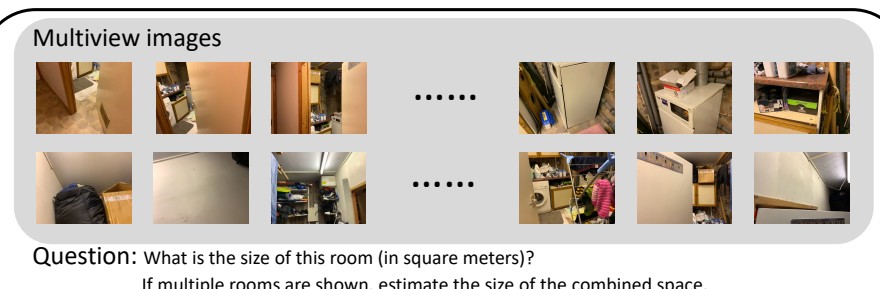

Multiview images

Question: What is the size of this room (in square meters)?
If multiple rooms are shown, estimate the size of the combined space.

Answer: **10.3**                                                    Ground Truth: **10.5**

(a) Results on room size estimation.

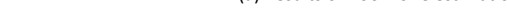

Multiview images

Question: What will be the first-time appearance order of the following categories in the video:
heater, tv, ceiling light, printer?
A. ceiling light, heater, tv, printer
B. ceiling light, tv, heater, printer
C. heater, tv, ceiling light, printer
D. ceiling light, heater, printer, tv

Answer: **B**

(b) Results on appearance order.

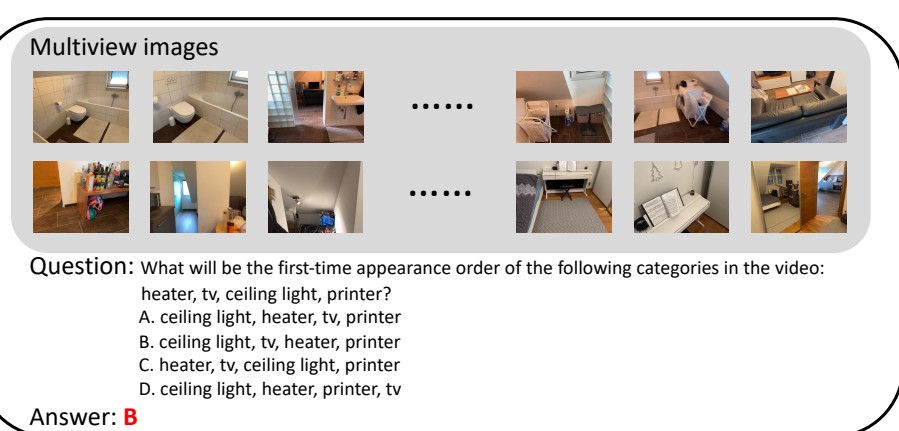

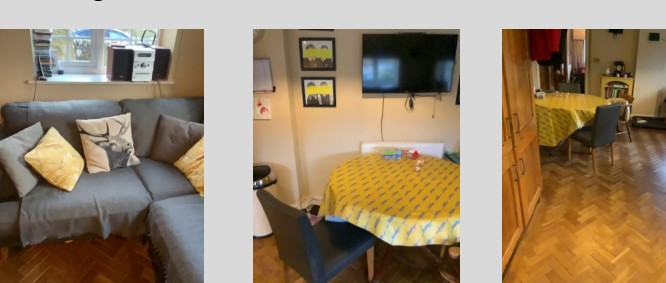

Multiview images

Question: Measuring from the closest point of each object,
which of these objects (stove, tv, table, sofa) is the closest to the stool?
Options:
A. stove
B. tv
C. table
D. sofa
Answer with the option's letter from the given choices directly.

Answer: **D.**                                                    GT: D.

(c) Object relative distance.

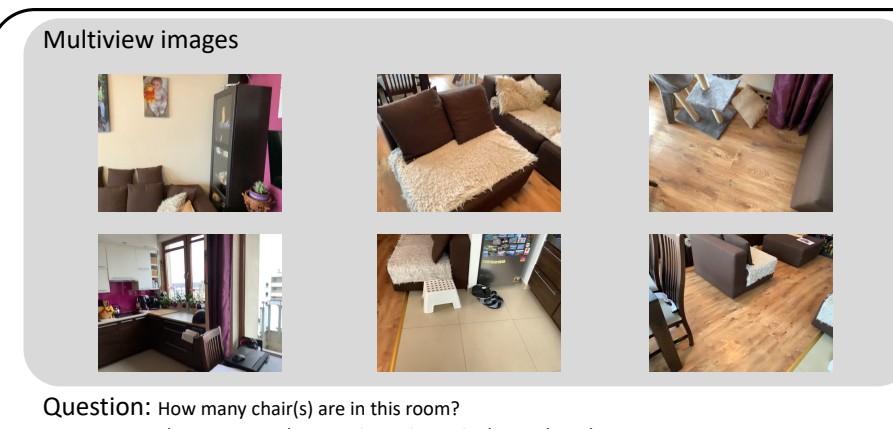

Question: How many chair(s) are in this room?
Please answer the question using a single word or phrase.

Answer: **4.**                                                          GT: 4.

(d) Object counting.

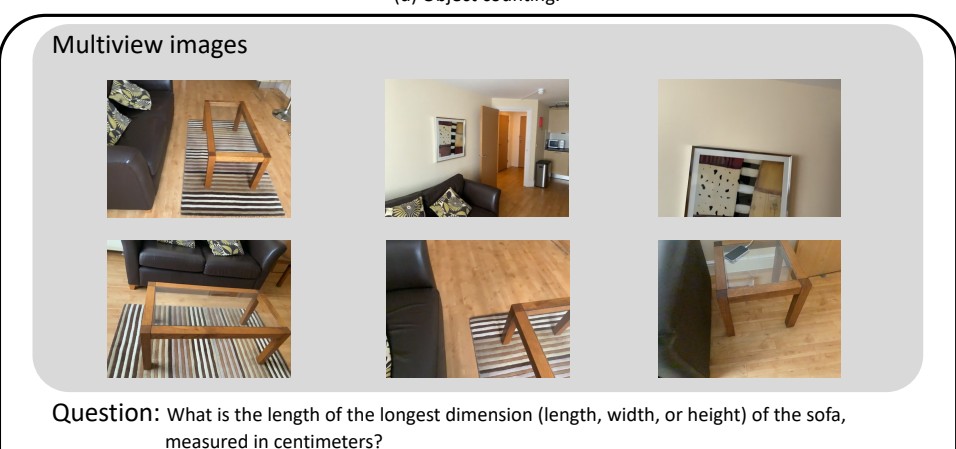

Question: What is the length of the longest dimension (length, width, or height) of the sofa, measured in centimeters?
Please answer the question using a single word or phrase.

Answer: **184.**                                                        GT: 173.

(e) Object size estimation.

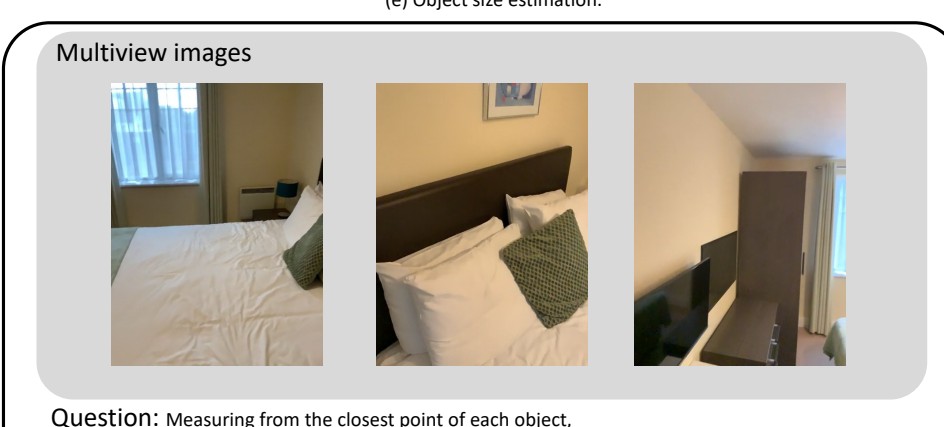

Question: Measuring from the closest point of each object, what is the distance between the tv and the bed (in meters)?
Please answer the question using a single word or phrase.

Answer: **1.0.**                                                        GT: 1.1.

(f) Object absolute distance.

Multiview images

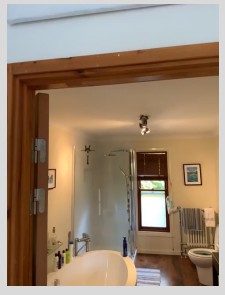 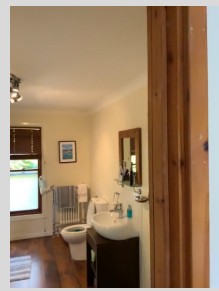 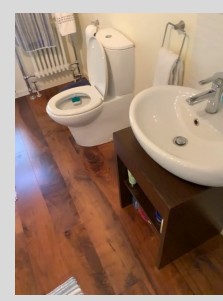

Question: If I am standing by the bathtub and facing the toilet,
is the table to my front-left, front-right, back-left, or back-right?
The directions refer to the quadrants of a Cartesian plane
(if I am standing at the origin and facing along the positive y-axis).
Options:
A. back-left
B. front-right
C. front-left
D. back-right
Answer with the option's letter from the given choices directly.

Answer: **B.**                                          GT: B.

(g) Object relative direction.

Multiview images

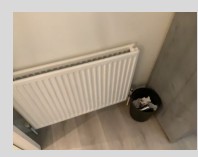 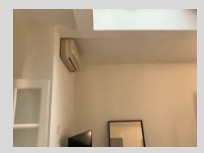 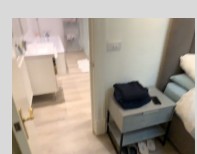
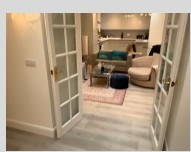 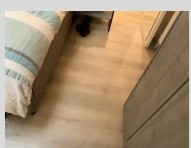 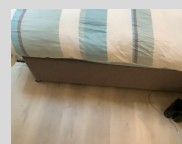

Question: You are a robot beginning at the tv facing the bed.
You want to navigate to the trash bin.
You will perform the following actions
(Note: for each [please fill in], choose either 'turn back,' 'turn left,' or 'turn right.'):
1. [please fill in]
2. Go forward until the cabinet
3. [please fill in]
4. Go forward until the trash bin is on your right.
You have reached the final destination.
Options:
A. Turn Left, Turn Left
B. Turn Right, Turn Left
C. Turn Back, Turn Left
D. Turn Right, Turn Right
Answer with the option's letter from the given choices directly.

Answer: **B.**                                          GT: B.

(h) Route planning.

## A.3 Scale consistency across scenes

We analyze the scale consistency across different scenes. To this end, we selected some test samples from the SPAR-7M dataset and used Omni-View to perform absolute distance prediction based on multi-view images. This SPAR-7M consists of scenes from three datasets: ScanNet and ScanNet++, where the average scene depth ranges from 3 to 5 meters with a maximum depth of 12 meters; and Structured3D, which features larger-scale environments with average depths between 4 and 6 meters and maximum depths reaching up to 20 meters. Qualitative visualizations demonstrate that Omni-View can predict absolute depth in scenes with different metric scale.

Multiview images

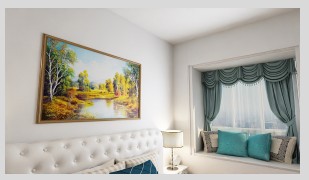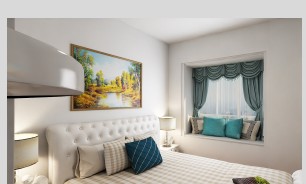

Question: The Object63 (red point at Frame-0, [0.14, 0.67]) is positioned at 3.4 meters depth.
Predict the depth of the Object22 (blue point at Frame-0, [0.23, 0.65]).
Calculate or judge based on the 3D center points of these objects.

Answer: **Object22's central depth is estimated to be about 2.0 meters.**

GT: With a central depth of 2.2 meters, Object22 is referenced here.

(a) Absolute depth estimation in structured3d, scene_03201_80.

Multiview images

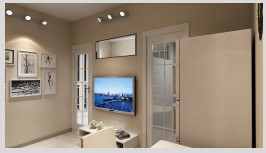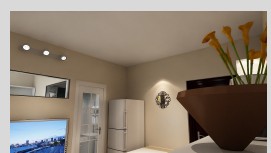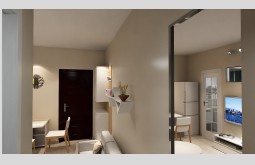

Question: The Object26 (at Frame-1, [0.10, 0.39]) lies at 2.7 meters depth.
Predict the depth of the Object30 (at Frame-1, [0.10, 0.70]).
Calculate or judge based on the 3D center points of these objects.

Answer: **Object30 is centered at about 4.0 meters depth.**

GT: At its center, object Object30 has a depth of 4.3 meters.

(b) Absolute depth estimation in structured3d, scene_02053_25.

Multiview images

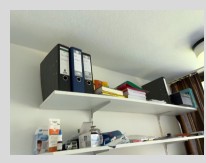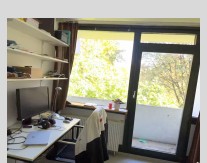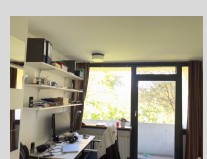

Question: If the plant (Frame-1, [0.30, 0.41]) is positioned at 2.7 meters,
estimate the depth of the laptop (at Frame-1, [0.16, 0.63]).
Calculate or judge based on the 3D center points of these objects.
The answer should be a single number, nothing more. .

Answer: **2.4**                                    GT: 2.3

(c) Absolute depth estimation in scannetpp, 9d7e20fbec_6022.

## A.4 NVS FROM SINGLE VIEW

**Indoor scenes.** We present the two visual results of NVS in indoor scenes. Omni-View can reasonably imagine unseen areas while maintaining the texture and structure of observed objects, like the followers on the table in the first row.

| Reference | View-12 | View-24 | View-36 | View-48 |

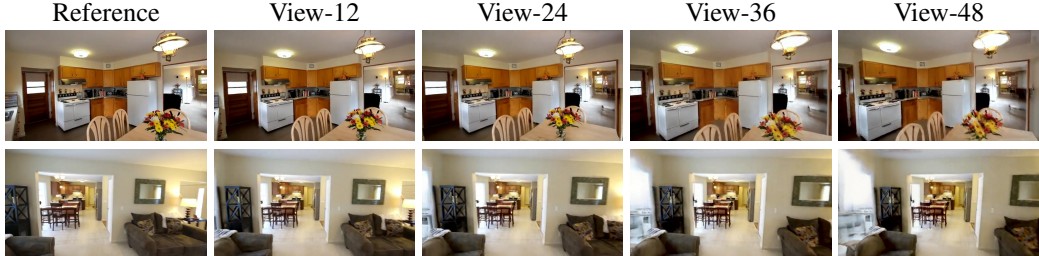

**Outdoor scenes.** We present the two visual results of NVS in outdoor scenes. When the camera movement is small, Omni-View can consistently generate new views.

| Reference | View-8 | View-16 | View-24 | View-32 |

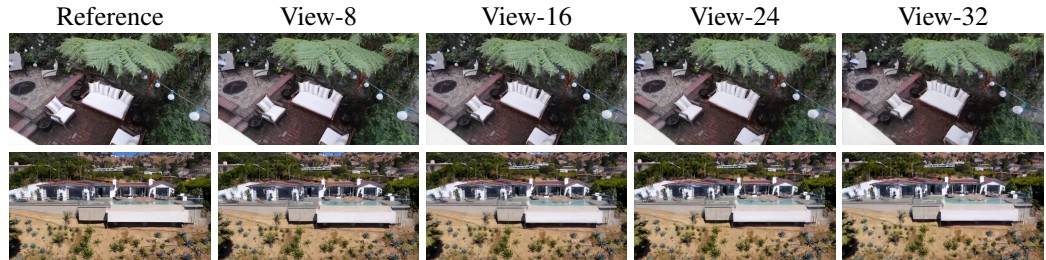

**Depth estimation.** We present the depth prediction results of Omni-View in indoor and outdoor scenes. In indoor scenes, the output of Omni-View's geometry module is more accurate.

| Reference | View-10 | View-20 | View-30 | View-40 |

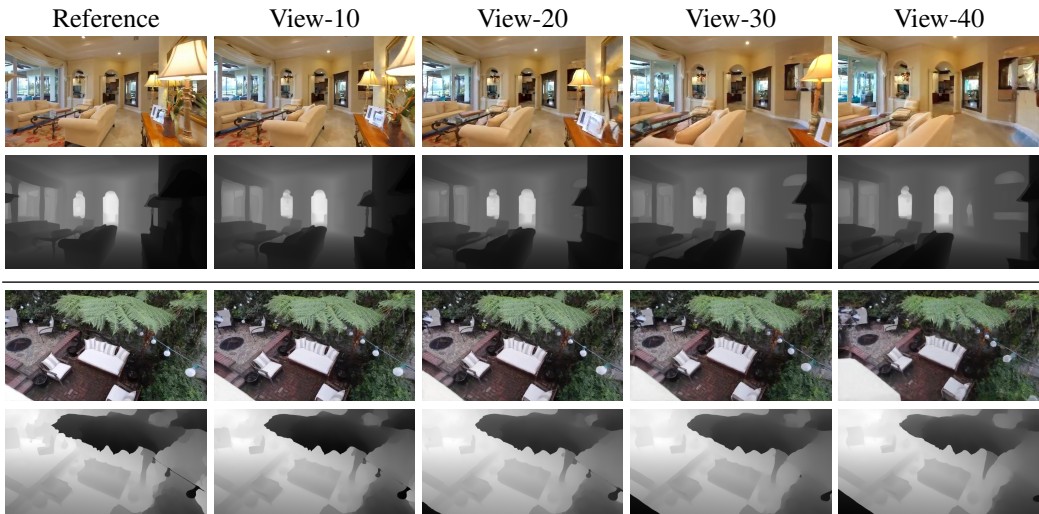

**Failure case.** We claim that the existing Omni-View is inadequate for effectively processing outdoor scenes with substantial camera movement. As illustrated in the figure below, the result images from Omni-View exhibit significant artifacts, highlighted within the red dashed boxes. Future work will focus on resolving the following challenges to enhance the handling of outdoor scenes with extensive camera motion: (1) the development of more precise camera control mechanisms, and (2) the improvement of inter-frame texture consistency stability.

| Reference | View-2 | View-4 | View-6 | View-8 |

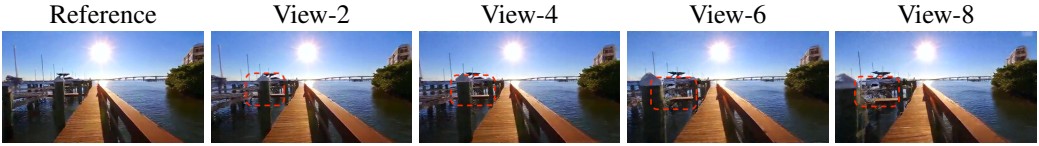

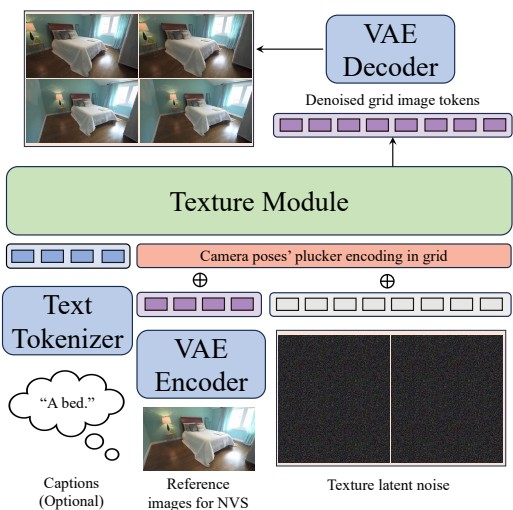

Figure 2: Generation in grid.

## A.5 EFFECT OF GENERATION IN GRID

**Generation in grid.** We explored small improvements to enhance inter-frame consistency and support long-sequence scene generation: "generation in grid" Hu et al. (2026). Specifically, we arrange 4 consecutive views into a single frame by organizing them in a grid layout, and perform autoregressive generation over the resulting sequence of grid-organized frames.

**Results.** As shown in the figures on the next page, we demonstrate that after using "generation in grid", Omni-View can generate more consistent novel view images.

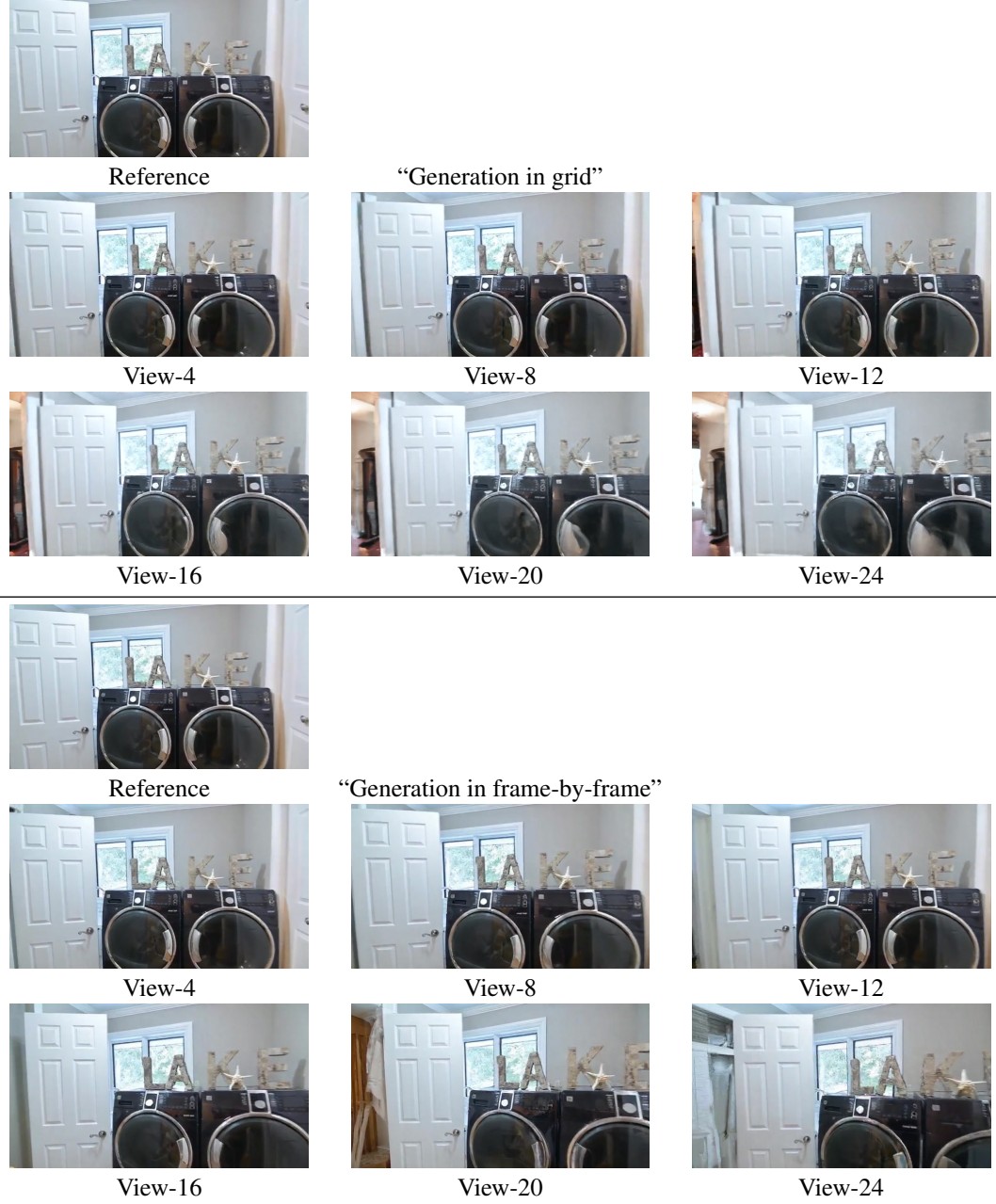

Figure 3: Different inference strategy in NVS (RealEstate10k, e7020d449f50c737). The first three rows use the generation in grid strategy, achieving significant improvements in inter-frame consistency. The last three rows still use the frame-by-frame generation strategy, which has poor inter-frame consistency.

# B TECHNICAL DETAILS

## B.1 DATASET

**3D scene understanding and spatial reasoning.** We curate a filtered training dataset containing 780k valid samples by consolidating data from multiple sources, including SQA3D, ScanQA, 3DOD, ScanRefer, VLM-3R, a 234k subset of SPAR from VG-LLM, and a 64k subset of llava-hound4 from VG-LLM. To ensure data quality, we perform two main filtering steps: (i) deduplication across the combined dataset, and (ii) removal of samples with invalid bounding box annotations. Specifically, we convert all bounding boxes to the $[x_1, y_1, x_2, y_2]$ format and exclude any instance where $x_1 < 0$, $y_1 < 0$, $x_2 >$ width, or $y_2 >$ height.

**Novel view synthesis.** We select 61k video clips from RealE10K train set, excluding those with fewer than 32 frames or significant motion blur that could degrade training stability. Depth maps are synthesized using the Voyager data pipeline, while captions are generated via QwenVLMax.

## B.2 ARCHITECTURE.

**Understanding model.** The detailed architecture is shown in Figure 4. The text tokenizer inherits from the vocabulary used by Qwen2. The image tokenizer for understanding is SigLIP. The backbone has layers as 28, attention head as 28, and hidden size as 3584, totaling 7B parameters. Key architectural choices include SwiGLU as the activation function, RMSNorm for normalization, FlexAttention (via PyTorch) for efficient self-attention computation, and position encoding adapted from Qwen2.

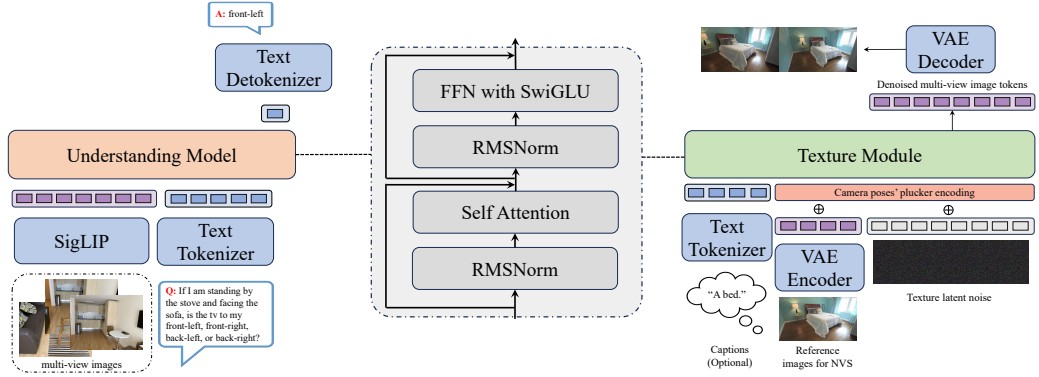

Figure 4: Architecture of understanding model and texture module.

**Texture module.** This module uses FLUX-VAE as the image tokenizer, sharing the same backbone architecture as the understanding model.

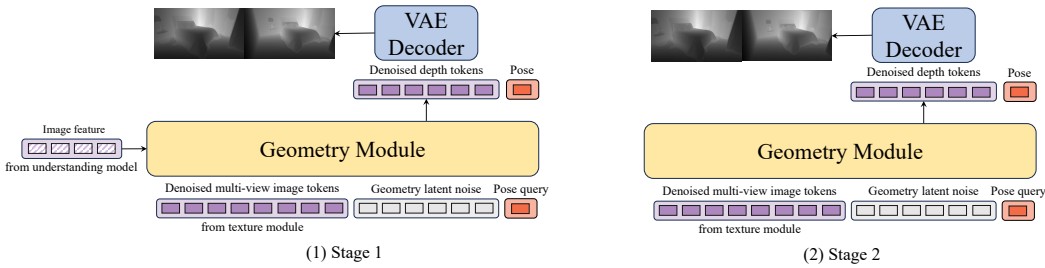

(1) Stage 1                    (2) Stage 2

Figure 5: Architecture of geometry module in two stages.

**Geometry module.** The detailed architecture is shown in Figure 5. Depth maps are also encoded by FLUX-VAE. Its backbone is basically the same as the block architecture, but reduced to 4 layers, containing about 1B parameters. Unlike the others, the architecture of the geometry module changes across different training stages. In stage 1, it receives features from the understanding model for

cross-attention. In stage 2, it no longer receives outputs from the understanding model. The reason for this design can be found in the discussion of stage 2 in Section 3.2.

### B.3    TRAINING AND COMPUTATIONAL COST.

All training phases use the AdamW optimizer with $\beta_1 = 0.9, \beta_2 = 0.95$, a peak learning rate of $1 \times 10^{-5}$, and a linear warm-up schedule covering the first 5% of iterations.

**Stage 1.** We train on the 3D scene understanding dataset for 10,000 iterations with a packed sequence length of 50k, using 32 H100 GPUs, which takes approximately 160 hours.

**Stage 2.** For the generation task, we perform 20,000 iterations on the novel view synthesis dataset with a packed sequence length of 32k, using 32 H100 GPUs, requiring approximately 40 hours in total.

**Understanding inference.** The model takes approximately 2.5 seconds on average to process a 32-frame multi-view scene understanding query using a single H100 GPU.

**Generation inference.** Generating a single image at resolution $640 \times 352$ takes about 2.2 seconds on average with one H100 GPU.

### B.4    CONVERGENCE BEHAVIOR.

**Stage 1.** Overall, most losses exhibit smooth and stable convergence. However, we observe spikes in the camera pose loss during training, particularly in later epochs. We hypothesize that this behavior may stem from instable optimizion by learnable queries in the camera pose estimation task. However, these fluctuations do not prevent the model from converging to effective solutions, as evidenced by strong performance in 3D scene understanding, spatial reasoning, and novel view generation.

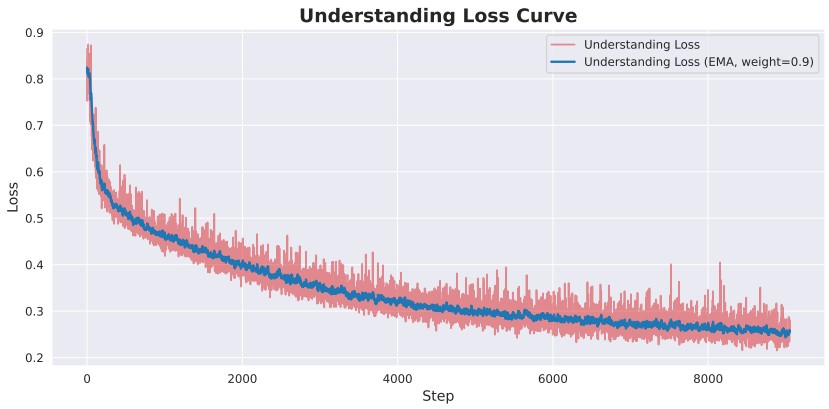

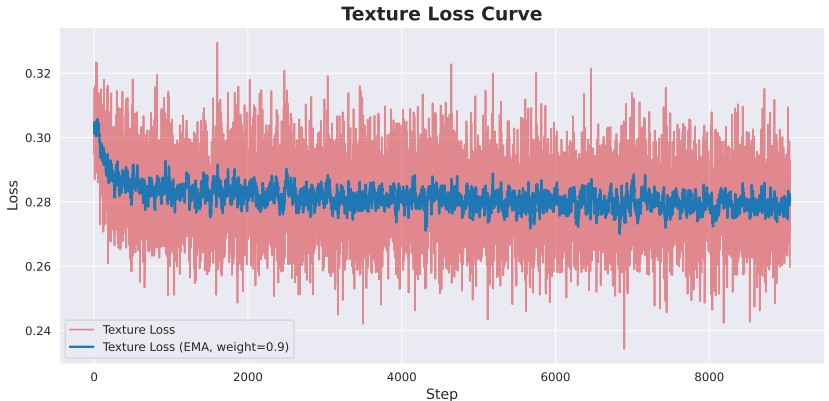

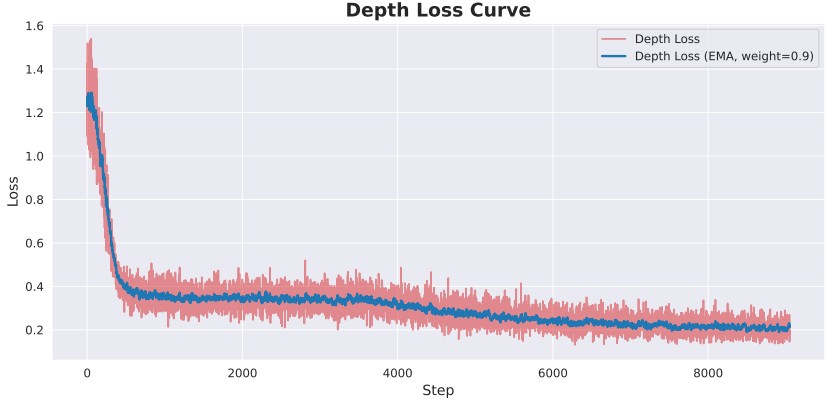

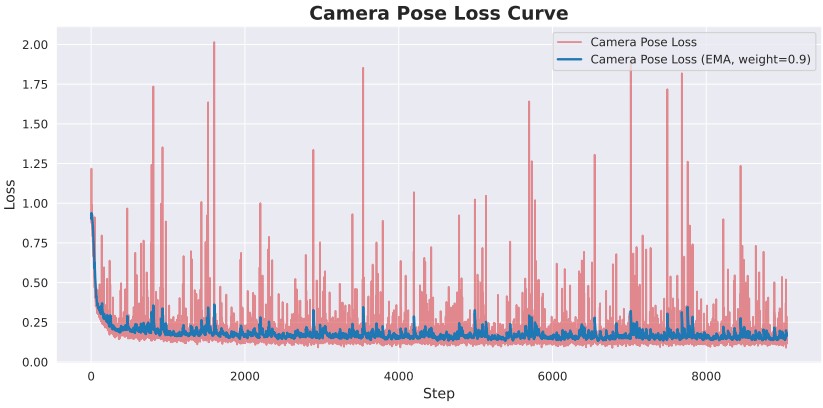

**Stage 2.** Since we used a grid generation format in stage 2, the per-image prediction method in stage 1 cannot be directly transferred to the "generation in grid" paradigm in stage 2. However, due to the generative prior learned in stage 1, Omni-View can quickly converge to a lower loss in stage 2. However, it was observed that the geometry loss showed a slight increase at around 8000 iterations, followed by a decrease, suggesting that it might be possible to further reduce $\lambda_{geo}$ to improve the training stability.

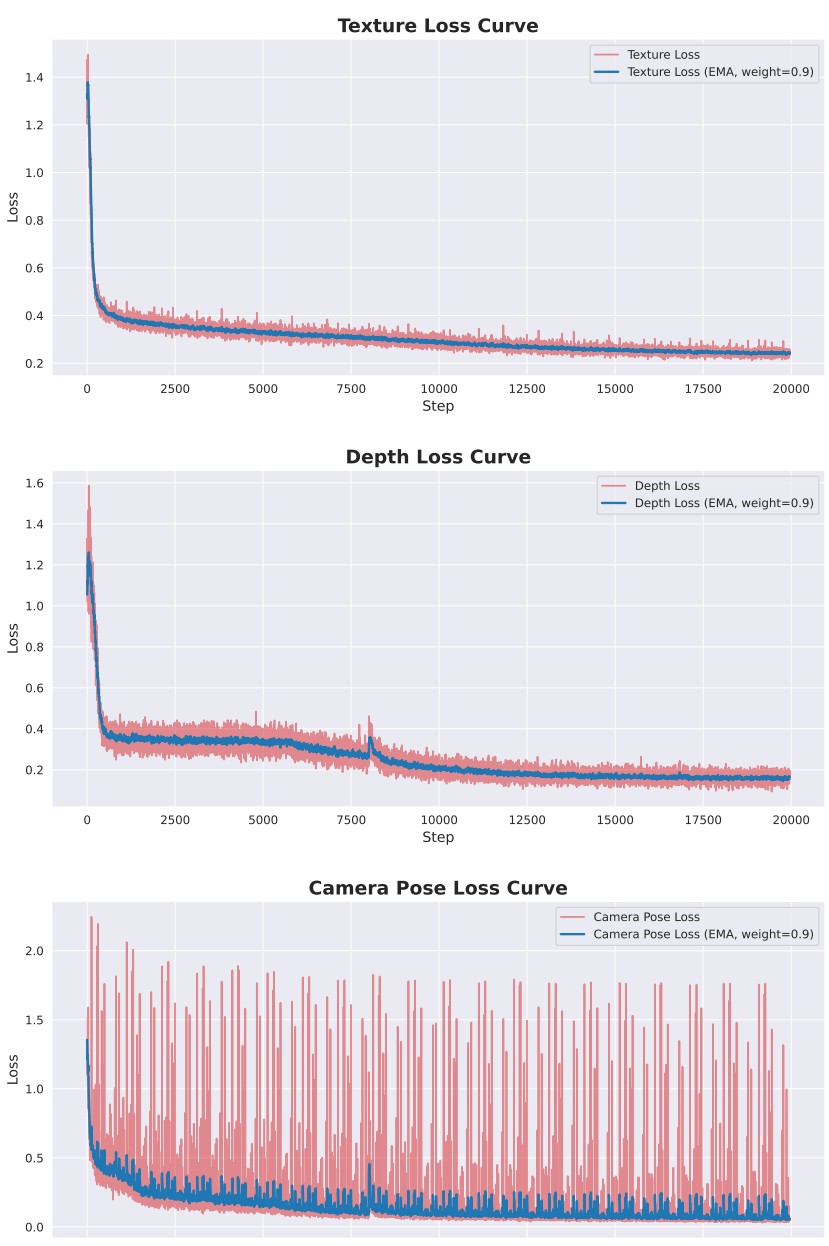

### B.5 Activation visualization

We visualize the activation maps of Bagel and Omni-View when they perform spatial reasoning tasks, as shown in the figure below. In this example, we want the model to locate how many cabinets are in the current scene, using the prompt: "`How many cabinet(s) are in this room?`". It can be seen that Bagel mainly focuses on the first two images that are irrelevant to

the question, whereas Omni-View is able to attend to each image as much as possible. The wider attention may be the reason why Omni-View performs better on 3D scene understanding tasks.

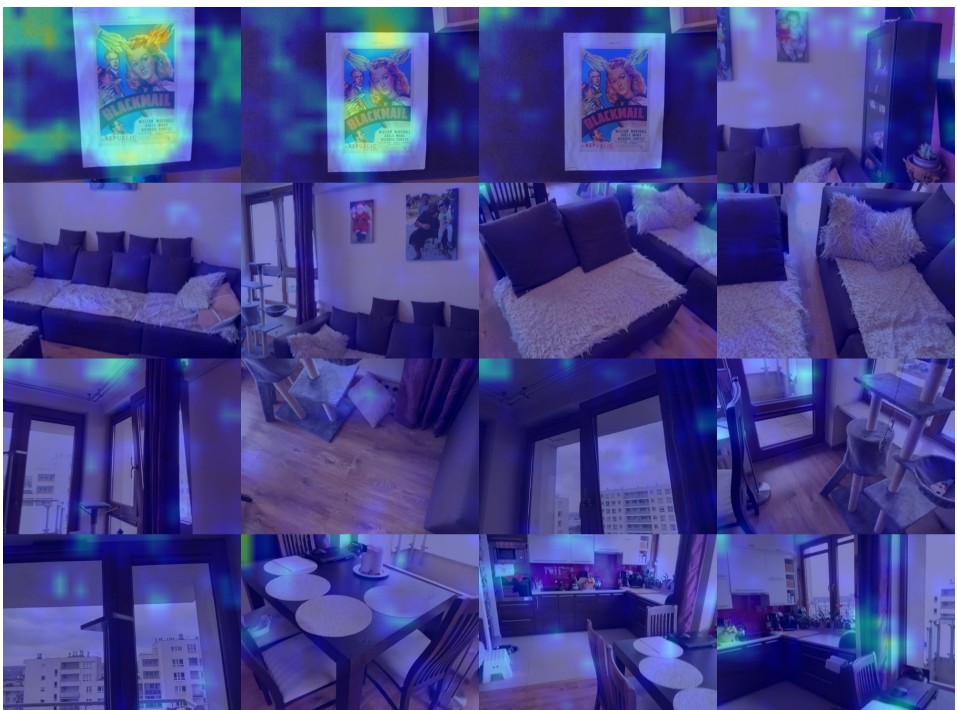

Bagel-FT

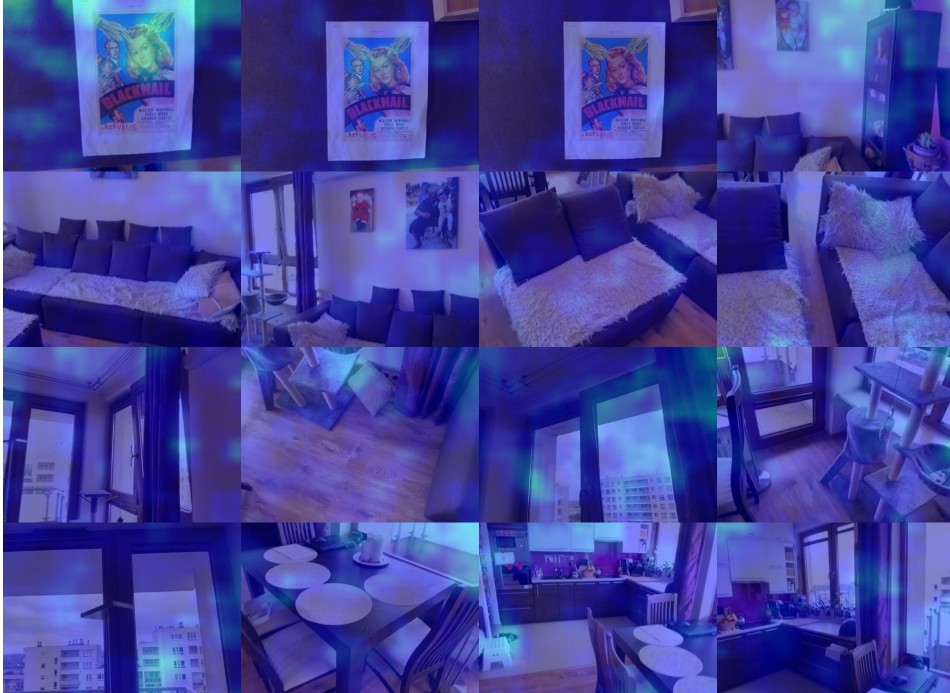

Omni-View (Ours)

Figure 6: Activation map. Bagel-FT vs. Omni-View.

## C  THE USE OF LARGE LANGUAGE MODELS (LLMS)

LLMs are used to correct potential grammatical inaccuracies in the manuscript. LLMs do not participate in research ideation.

