# OpenReview forum: "Omni-View: Unlocking How Generation Facilitates Understanding in Unified 3D Model based on Multiview images"
_ICLR.cc/2026/Conference — ICLR 2026 Poster_

### Official Review · Reviewer_GSip · 2025-10-28

**Soundness:** 3
**Presentation:** 3
**Contribution:** 3
**Rating:** 6
**Confidence:** 3

**Summary:**

This paper proposes Omni-View, a unified 3D scene understanding and generation model based on multi-view imagery, exploring the principle of "generation promotes understanding." The model, composed of an understanding module, a texture module, and a geometry module, jointly models scene understanding, novel view synthesis, and geometry estimation. Using a two-stage training strategy, Omni-View achieved a state-of-the-art score of 55.4 on the VSI-Bench, surpassing existing dedicated 3D understanding models while also performing well in novel view synthesis and scene generation tasks.

**Strengths:**

Unified modeling is novel: This is the first systematic exploration of the "generation-driven understanding" mechanism in 3D scenes, which is both inspiring and forward-looking.

Rational modular design: The generation module is split into texture and geometry components, modeling appearance and structure respectively, effectively improving understanding capabilities.

Effective training strategy: The two-stage training (unified training + generation fine-tuning) balances understanding and generation performance, and the D2S mechanism improves robustness.

Comprehensive experiments: The effectiveness of the method is verified on multiple 3D understanding, spatial reasoning, and generation tasks, with results significantly outperforming existing unified models.

No 3D input required: Relying solely on multi-view images, this improves the model's practicality and generalization capabilities.

**Weaknesses:**

Weak theoretical analysis: Although "generation promotes understanding" has been proposed, there is a lack of theoretical or interpretable analysis of its underlying mechanisms.

Generation quality still has room for improvement: Despite leading in PSNR/SSIM, inter-frame consistency under large viewpoint variations remains suboptimal (see Appendix visualization).

The geometry module relies on synthetic data: The depth map is synthesized by Voyager, which may limit the realism and accuracy of geometry predictions.

Limited long sequence generation capability: The model currently does not support long sequence scene generation, limiting its application in open-world scenarios.

There is still a gap compared to state-of-the-art dedicated models: In particular, in the 3D grounding task, there is still a significant gap compared to methods that rely on 3D input.Strengthen theoretical analysis or visual explanation of the "generation promotes understanding" mechanism;

Compare with state-of-the-art methods on more 3D grounding tasks and analyze the sources of the gap.

**Questions:**

Strengthen theoretical analysis or visual explanation of the "generation promotes understanding" mechanism;

Compare with state-of-the-art methods on more 3D grounding tasks and analyze the sources of the gap.

---

> ### Author Response · Authors · 2025-11-20
>
> Thanks for your insightful feedback and your time in reading our paper. We first list your advice and questions, then give our detailed answers.
>
> > Weakness 1 & Question 1. Weak theoretical analysis: Although "generation promotes understanding" has been proposed, there is a lack of theoretical or interpretable analysis of its underlying mechanisms. Strengthen theoretical analysis or visual explanation of the "generation promotes understanding" mechanism.
>
> **Answer.** We appreciate your insightful suggestion. In this work, we attempt to investigate the underlying mechanisms of "generation promotes understanding" through visualization [1]. The Figure 6 in the revised Appendix.B.5 shows that Omni-View is able to attend to each image as much as possible. This broader attention pattern may contribute to model's improved performance on 3D scene understanding tasks. We believe this is a promising direction and plan to further explore the underlying reasons behind "generation promotes understanding" in future work.
>
> ---
>
> [1] Token Activation Map to Visually Explain Multimodal LLMs. ICCV 2025.

---

> ### Author Response · Authors · 2025-11-20
>
> > Weakness 2 & Weakness 4. Generation quality still has room for improvement: Despite leading in PSNR/SSIM, inter-frame consistency under large viewpoint variations remains suboptimal (see Appendix visualization). Limited long sequence generation capability: The model currently does not support long sequence scene generation, limiting its application in open-world scenarios.
>
> **Answer.** The weaknesses you raise are indeed important issues in our paper, but they can be addressed. We explored a small improvement to enhance inter-frame consistency and support long-sequence scene generation: **generation in grid**.
>
> Specifically, we arrange 4 consecutive views into a single frame by organizing them in a grid layout, and perform autoregressive generation over the resulting sequence of grid-organized frames. We retrained stage 2 under this "generation in grid" strategy and obtained the following results.
>
> | Method | PSNR | SSIM | LPIPS |
> | -- | -- | -- | -- |
> | ViewCrafter | 22.60 | 0.754 |  0.195 |
> | Voyager | 23.12 | 0.793 | 0.175 |
> | Omni-View (no grid) | 23.22 | 0.817 | 0.114 |
> | Omni-View (grid) | 23.30 | 0.823 | 0.111 |
>
> - *Quantitative results.* As shown in the above table, With this "generation in grid" strategy, our Omni-View achieves better performance on the novel view synthesis (NVS) task.
> - *Qualitative results.* We present qualitative examples in the Appendix A.4 and Appendix A.5 showcasing this approach across indoor and outdoor scenes. The qualitative results demonstrate that Omni-View is able to support generation of longer sequences, extending the length from 25 to 50, and achieves improved consistency across views when using this "generation in grid" strategy.
>
> Meanwhile, *we will explore more solutions in the future*, such as memory retrieval (context-as-memory [1]), introducing better training methods (self forcing [2]), and reinforcement learning (FlowGRPO [3]). We consider enhancing Omni-View's scene video generation capabilities an important future work.
>
> ---
>
> [1] Context as memory: Scene-consistent interactive long video generation with memory retrieval. Siggraph 2025.
>
> [2] Self Forcing: Bridging the Train-Test Gap in Autoregressive Video Diffusion. NeurIPS 2025.
>
> [3] Flow-GRPO: Training Flow Matching Models via Online RL. NeurIPS 2025.

---

> ### Author Response · Authors · 2025-11-20
>
> > Weakness 3. The geometry module relies on synthetic data: The depth map is synthesized by Voyager, which may limit the realism and accuracy of geometry predictions.
>
> **Answer.** Thank you for your valuable feedback regarding the realism and accuracy of the geometry predictions. We would first like to explain why we're using simulated data and discuss how to achieve realism and accuracy in geometry predictions.
>
> We use simulated data for the following reasons:
>
> 1. **Availability of large-scale accurate depth data.** Datasets with precise depth annotations are significantly more limited in scale compared to widely available monocular video collections.
> 2. **Primary motivation of the geometry module.** The goal of the geometry module in Omni-View is not to achieve state-of-the-art depth prediction accuracy, but rather to learn meaningful 3D structural representations that support improved 3D scene understanding.
>
> To further support this choice, we conduct geometric prediction visualizations on **both indoor and outdoor scenes**, we present depth map visualizations on both indoor and outdoor scenes in the revised Appendix A.4.
>
> - **Depth maps in indoor scenes.** The visualizations show that Omni-View produces reasonable depth estimation in indoor scenes, consistent with the spatial layouts.
> - **Generalization to outdoor scenes.** Despite being trained almost exclusively on indoor data from RealEstate10K, Omni-View is able to generate plausible depth maps for outdoor scenes. This cross-domain generalization suggests that the geometry module has learned underlying 3D structural priors rather than simply memorizing domain-specific patterns.
>
> These results indicate that the model captures meaningful 3D knowledge, which contributes to its overall scene understanding capability.
>
> ---
>
> **Finally,** we also acknowledge that further improving geometric fidelity remains an important direction for future work. To improve the realism and accuracy of geometry predictions, we can use real data for training. However, the required datasets are too large (CUT3R mentions needing 20TB of storage), and the computational resources are also enormous (e.g., VGGT requires four days of training with 128 H100 GPUs). We cannot provide relevant results during the rebuttal. We consider this an important future work. Your proposal will help Omni-View move towards a more generalized unify 3D model.

---

> ### Author Response · Authors · 2025-11-20
>
> > Weakness 5, 6 & Queation 2. Compare with state-of-the-art methods on more 3D grounding tasks and analyze the sources of the gap.
>
> **Answer.** We believe the performance gap in 3D grounding tasks primarily stems from Omni-View’s lack of 3D-aware inputs, such as explicit 3D data (e.g., point clouds or camera poses) or 3D-informed features like VGGT. This hypothesis is supported by our analysis of existing experimental results, summarized in the table below:
>
> | Methods | 3D input | VGGT input | Acc @ 0.25 | Acc @ 0.5 |
> | -- | -- | -- | -- | -- |
> | Video-3D-LLM | ✔ | ✘ | 58.1 | 51.7 |
> | Ross3D | ✔ | ✘ | 61.1 | 54.4 |
> | SPAR | ✘ | ✘ | 48.8 | 43.1 |
> | VG-LLM | ✘ | ✔ | 53.5 | 47.5 |
> | Omni-View | ✘ | ✘ | 50.8 | 45.0 |
>
> From this comparison, three observations can be made:
>
> - Omni-View outperforms methods that do not use any 3D-aware inputs (e.g., SPAR).
> - Omni-View's performance is slightly lower than methods that incorporate 3D-informed features such as VGGT (e.g., VG-LLM), suggesting that access to VGGT priors can provide a meaningful advantage.
> - Omni-View lags significantly behind methods that leverage explicit 3D inputs (e.g., Video-3D-LLM or Ross3D), which appear to offer the strongest benefit for 3D grounding.
>
> These results suggest that incorporating 3D-aware of 3D-informed signals generally improves 3D grounding performance. This raises the question: can Omni-View’s 3D visual grounding capability be further enhanced by introducing 3D-aware of 3D-informed inputs?
>
> To investigate, we conduct ablation studies by augmenting Omni-View with additional 3D-aware inputs following Video3DLLM or 3D-informed inputs following VG-LLM, then evaluate on the 3D scene understanding task. Results are as follows:
>
> | Methods | 3D input | VGGT input | Acc @ 0.25 | Acc @ 0.5 |
> | -- | -- | -- | -- | -- |
> | Omni-View | ✘ | ✘ | 50.8 | 45.0 |
> | Omni-View | ✘ | ✔ | 53.9 | 48.2 |
> | Omni-View | ✔ | ✘ | 58.2 | 50.4 |
>
> - When VGGT features are added into the input, Omni-View shows a moderate improvement in 3D visual grounding performance.
> - When camera poses are treated as position embedding following Video 3D-LLM, we observe a more significant performance gain.
>
> These findings confirm that integrating explicit 3D information benefits Omni-View in 3D visual grounding, consistent with trends observed in prior work.

---

> > ### Comment · Reviewer_GSip · 2025-11-28
> >
> > Thank you for the author's response, which was very detailed and resolved my doubts.

---

### Official Review · Reviewer_U2JG · 2025-10-31

**Soundness:** 3
**Presentation:** 3
**Contribution:** 2
**Rating:** 4
**Confidence:** 3

**Summary:**

This paper introduces Omni-View, a unified model for 3D scene understanding and generation that explicitly investigates the hypothesis that “generation facilitates understanding.” The contributions of the paper are:
1. Unified architecture integrating 3D scene understanding and generation, composed of three main components:
1.1 Understanding model (for spatial reasoning and QA)
1.2 Texture module (for novel view synthesis)
1.3 Geometry module (for depth and pose estimation)
2. Proposed a novel two-stage training strategy:
2.1 Jointly trains understanding and generation to encourage mutual benefits through geometry and spatiotemporal modeling.
2.2 Fine-tunes generation with RGB-Depth-Pose joint learning for better geometric consistency.
3. Empirical validation showing state-of-the-art (SOTA) performance on the VSI-Bench (score 55.4), outperforming both specialized and unified 3D models in reasoning tasks.
Overall, Omni-View demonstrates how generative modeling (novel view synthesis, geometry estimation) can enhance 3D reasoning, localization, and understanding—a conceptually elegant and empirically supported contribution.

**Strengths:**

1. Clear intuition and solid empirical validation.
The paper builds upon a clear and intuitive idea — that generation can facilitate understanding — and the overall logic is easy to follow. Quantitative results across multiple benchmarks convincingly demonstrate the benefits of the proposed design, especially in spatial reasoning and novel view synthesis.
2. Architectural innovation.
By decomposing the generation process into texture and geometry modules, the authors present a meaningful and modular architecture that captures both appearance and structure. This decomposition aligns well with human visual reasoning and can be viewed as an innovative contribution for the community.
3. Comprehensive ablation studies.
The ablation results thoroughly verify the contributions of the proposed contributions. These analyses effectively demonstrate that each component contributes to the final understanding and reasoning performance.

**Weaknesses:**

1. The qualitative results in the appendix are sparse, and there are no depth estimation visualizations or broader test cases. This makes it difficult to verify the model’s generalization and effectiveness beyond the reported metrics. For instance, the quality and consistency of metric-scale prediction from the geometry module remain uncertain — the reported results could be influenced by selective visualization or data bias, since the paper lacks convincing examples that demonstrate accurate geometric reasoning across varied real-world scenes.
2. The technical details provided for training, implementation, and comparison setups are relatively limited. Without clearer supplementary material (e.g., dataset statistics, architecture specifics, or convergence behavior), it is challenging to fully reproduce the reported results or assess robustness under different conditions.
3. The absence of released code or live demonstrations restricts the ability of other researchers to validate or extend this work. Although acceptable for review, the paper would be strengthened by open-sourcing its checkpoints or providing additional evaluation on long-range world generation and 3D visual grounding.

**Questions:**

1. Could the authors provide qualitative results of the geometry module’s depth estimation, camera pose estimation with metric-scale? Without any depth estimation results or broader test cases, it is difficult to assess whether the geometry module truly learns meaningful 3D structure rather than overfitting to training priors, without any generalizability.
2. How well does Omni-View generalize to unseen or real-world multi-view scenes, rather than the well captured ones in the appendix? Have the authors tested its performance to verify the robustness and effectiveness of the learned spatial reasoning?
3. Could the authors clarify key implementation and training details—such as dataset splits, optimizer configurations, training epochs, and computational cost—to ensure reproducibility? Including more specifics or releasing supplementary materials would improve transparency.

**Details Of Ethics Concerns:**

There is no ethics concern from the reviewer's side.

---

> ### Author Response · Authors · 2025-11-20
>
> Thank you for your insightful and comprehensive feedback. We have carefully studied your comments and argue that **your concerns can be addressed**.
>
> > Weakness 1 & Question 1. The qualitative results in the appendix are sparse, and there are no depth estimation visualizations or broader test cases. This makes it difficult to verify the model’s generalization and effectiveness beyond the reported metrics. For instance, the quality and consistency of metric-scale prediction from the geometry module remain uncertain — the reported results could be influenced by selective visualization or data bias, since the paper lacks convincing examples that demonstrate accurate geometric reasoning across varied real-world scenes. Could the authors provide qualitative results of the geometry module’s depth estimation, camera pose estimation with metric-scale? Without any depth estimation results or broader test cases, it is difficult to assess whether the geometry module truly learns meaningful 3D structure rather than overfitting to training priors, without any generalizability.
>
> A. Thank you for your suggestion. **We provide visualizations results related to geometry estimation** to demonstrate that our model's geometry module learns meaningful 3D structures with generalization rather than overfitting to training priors.
>
> To further support this point, we conduct geometric prediction evaluations on **both indoor and outdoor scenes**, we present depth map visualizations on both indoor and outdoor scenes in the revised Appendix A.4.
>
> Specifically,
>
> - **Depth maps in indoor scenes.** The visualizations show that Omni-View produces reasonable depth estimation in indoor scenes, consistent with the spatial layouts.
> - **Generalization to outdoor scenes.** Despite being trained almost exclusively on indoor data from RealEstate10K, Omni-View is able to generate plausible depth maps for outdoor scenes. This cross-domain generalization suggests that the geometry module has learned underlying 3D structural priors rather than simply memorizing domain-specific patterns.
> - **Camera pose estimation.** Since datasets such as Re10k and Tanks & Temples provide accurate camera poses and since camera poses themselves are difficult to visualize directly, we choose to present quantitative results for the predicted camera poses.
>
> We evaluate the camera pose accuracy of Omni-View on video datasets with accurate camera poses: Re10k and Tanks & Temples. Among these, Tanks & Temples consist of outdoor scenes, with Tanks & Temples featuring larger camera motions. We evaluate them using the standard metric AUC@30, which combines RRA and RTA. RRA (Relative Rotation Accuracy) and RTA (Relative Translation Accuracy) calculate the relative angular errors in rotation and
> translation, respectively.
>
> | Dataset | Re10k | Tanks & Temples |
> | -- | -- | -- |
> | VGGT | 85.3 | 87.8 |
> | Omni-View | 85.6 | 82.5 |
>
> As shown in the results, Omni-View demonstrates reasonable generalization to outdoor scenes and scenes with diverse camera motions, despite being primarily trained on indoor scenes. However, its camera pose estimation accuracy is weaker compared to 3D reconstruction model, like VGGT.
>
> ---
>
> **Finally,** we would like to clarify that the performance gap in geometry estimation is expected, as **the motivation for designing the geometry module was to investigate whether jointly learning a geometry estimation task could enhance scene understanding performance**, rather than to achieve state-of-the-art accuracy on the geometry estimation task itself. This explains why we did not include evaluation on geometry estimation tasks in the initial version of the paper.

---

> ### Author Response · Authors · 2025-11-20
>
> > Weakness 2 & Question 3. The technical details provided for training, implementation, and comparison setups are relatively limited. Without clearer supplementary material (e.g., dataset statistics, architecture specifics, or convergence behavior), it is challenging to fully reproduce the reported results or assess robustness under different conditions. Could the authors clarify key implementation and training details—such as dataset splits, optimizer configurations, training epochs, and computational cost—to ensure reproducibility? Including more specifics or releasing supplementary materials would improve transparency.
>
> **Answer.** Thank you for your valuable feedback. We are willing to provide additional technical details to further improve the reproducibility of Omni-View. Below, we present a structured description of the technical details of Omni-View across four key aspects: dataset, architecture, training and computational cost, and convergence behavior.
>
> **(1/4) Dataset.**
> - *3D scene understanding and spatial reasoning.* We curate a filtered training dataset containing 780k valid samples by consolidating data from multiple sources, including SQA3D, ScanQA, 3DOD, ScanRefer, VLM-3R, a 234k subset of SPAR from VG-LLM, and a 64k subset of llava-hound4 from VG-LLM. To ensure data quality, we perform two main filtering steps: (i) deduplication across the combined dataset, and (ii) removal of samples with invalid bounding box annotations. Specifically, we convert all bounding boxes to the $[x_1, y_1, x_2, y_2]$ format and exclude any instance where $x_1 < 0$, $y_1 < 0$, $x_2 > \text{width}$, or $y_2 > \text{height}$.
> - *Novel view synthesis.* We select 61k video clips from RealE10K train set, excluding those with fewer than 32 frames or significant motion blur that could degrade training stability. Depth maps are synthesized using the Voyager data pipeline, while captions are generated via QwenVLMax.
>
> ---
>
> **(2/4) Architecture.**
>
> - *Understanding model.* The text tokenizer inherits from the vocabulary used by Qwen2. The image tokenizer for understanding is SigLIP. The backbone has layers as 28, attention head as 28, and hidden size as 3584, totaling 7B parameters. Key architectural choices include SwiGLU as the activation function, RMSNorm for normalization, FlexAttention (via PyTorch) for efficient self-attention computation, and position encoding adapted from Qwen2.
> - *Texture module.* This module uses FLUX-VAE as the image tokenizer, sharing the same backbone architecture as the understanding model.
> - *Geometry module.* Depth maps are also encoded by FLUX-VAE. Its backbone is basically the same as the block architecture, but reduced to 4 layers, containing about 1B parameters.
>
> ---
>
> **(3/4) Training and computational cost.**
>
> All training phases use the AdamW optimizer with $\beta_1 = 0.9, \beta_2 = 0.95$, a peak learning rate of $1 \times 10 ^{-5}$, and a linear warm-up schedule covering the first 5% of iterations.
>
> - *Training Stage 1:* We train on the 3D scene understanding dataset for 10,000 iterations with a packed sequence length of 50k, using 32 H100 GPUs, which takes approximately 160 hours.
> - *Training Stage 2:* For the generation task, we perform 20,000 iterations on the novel view synthesis dataset with a packed sequence length of 32k, using 32 H100 GPUs, requiring approximately 40 hours in total.
>
> - *Understanding inference.* The model takes approximately 2.5 seconds on average to process a 32-frame multi-view scene understanding query using a single H100 GPU.
> - *Generation inference.* Generating a single image at resolution 640 × 352 takes about 2.2 seconds on average with one H100 GPU.
>
> ---
>
> **(4/4) Convergence behavior.**
>
> In the revised Appendix B.4, we include convergence curves for all loss components in training stages. Overall, most losses exhibit smooth and stable convergence. However, we observe spikes in the camera pose loss during training, particularly in later epochs. We hypothesize that this behavior may stem from unstable optimization by learnable queries in the camera pose estimation task. However, these fluctuations do not prevent the model from converging to effective solutions, as evidenced by strong performance in 3D scene understanding, spatial reasoning, and novel view generation.
>
> ---
>
> **We hope these clarifications help enhance transparency and reproducibility.**

---

> ### Author Response · Authors · 2025-11-20
>
> > Weakness 3-1. The absence of released code or live demonstrations restricts the ability of other researchers to validate or extend this work.
>
> **Answer.** Due to the anonymity requirements, it's hard to release the model (more than 60G storage) for evaluation during the double-blind peer review. To mitigate the limitations caused by the inability to release runnable code, we prepared visualizations in revised Appendix A and supplementary that shows the model’s capabilities in 3D scene understanding, spatial reasoning, and novel view synthesis (NVS).
>
> - **3D scene understanding.** We provide visual examples in Appendix A.1, including tasks such as scene question answering and 3D object detection. For the detection task, we visualize the predicted 3D bounding boxes to demonstrate the accuracy of the estimated coordinates.
> - **Spatial reasoning.** We present qualitative results in Appendices A.2 and A.3, covering tasks in VSI-Bench. Omni-View supports multiple output formats, including free-form text, numerical values, and multiple-choice responses. Additionally, in Appendix A.3, we include an analysis of scene-consistency in absolute scale depth prediction task, showing that Omni-View is capable of estimating absolute depth even across scenes with different metric scales.
> - **NVS from a single view.** We present results in Appendices A.4 and the supplementary materials. Appendix A.4 showcases Omni-View’s performance in both indoor and outdoor scenes, along with several failure cases where significant camera motion poses challenges. The supplementary materials include the videos synthesized by Omni-View, all of them have been anonymized to protect privacy.
>
> We hope these additions provide a clearer and more comprehensive picture of Omni-View's capabilities, and we *will publicly release the code, datasets, and models upon publication*.

---

> ### Author Response · Authors · 2025-11-20
>
> > Weakness 3-2. Although acceptable for review, the paper would be strengthened by open-sourcing its checkpoints or providing additional evaluation on long-range world generation and 3D visual grounding.
>
> **Answer.** We analyzed the sources of performance limitations in long-range world generation and 3D visual grounding tasks, and have conducted preliminary explorations to identify potential directions for improvement.
>
> **(1/2) long-range world generation.** This weakness you raise is indeed an important issue in our Omni-View, but we believe it can be effectively mitigated.
>
> We attribute this weakness to the previously adopted per-image generation strategy. When generating a 50-frame sequence, the model performs 50 sequential autoregressive steps, which may lead to compounding errors over 50 times. In contrast, if we generate frames in chunks (e.g. 4 images per chunk), the total number of autoregressive steps reduces significantly (e.g., from 50 to 13), thereby reducing the opportunity for error propagation.
>
> Motivated by this insight, we explored a small improvement to enhance inter-frame consistency and support long-sequence scene generation: **generation in grid**. Specifically, we arrange 4 consecutive views into a single frame by organizing them in a grid layout, and perform autoregressive generation over the resulting sequence of grid-organized frames. We retrained stage 2 under this "generation in grid" strategy and obtained the following results.
>
> | Method | PSNR | SSIM | LPIPS |
> | -- | -- | -- | -- |
> | ViewCrafter | 22.60 | 0.754 |  0.195 |
> | Voyager | 23.12 | 0.793 | 0.175 |
> | Omni-View (no grid) | 23.22 | 0.817 | 0.114 |
> | Omni-View (grid) | 23.30 | 0.823 | 0.111 |
>
> - *Quantitative results.* As shown in the above table, With "generation in grid" strategy, Omni-View achieves better performance on the novel view synthesis (NVS) task.
> - *Qualitative results.* We present qualitative comparison between "generation in grid" and "generation in frame-by-frame" in the Appendix A.5. It demonstrates that Omni-View achieves improved inter-frame consistency when using this "generation in grid" strategy.
>
> Meanwhile, we will explore more solutions in the future, such as memory retrieval (context-as-memory [1]), better training methods (self forcing [2]), and reinforcement learning (FlowGRPO [3]). We consider enhancing Omni-View's scene video generation capabilities an important future work.
>
> ---
>
> **(2/2) 3D visual grounding.** We believe the performance gap in 3D grounding tasks primarily stems from Omni-View’s lack of 3D-aware inputs, such as explicit 3D data (e.g., point clouds or camera poses) or 3D-informed features like VGGT. This hypothesis is supported by our analysis of existing experimental results, summarized in the table below:
>
> | Methods | 3D input | VGGT input | Acc @ 0.25 | Acc @ 0.5 |
> | -- | -- | -- | -- | -- |
> | Video-3D-LLM | ✔ | ✘ | 58.1 | 51.7 |
> | Ross3D | ✔ | ✘ | 61.1 | 54.4 |
> | SPAR | ✘ | ✘ | 48.8 | 43.1 |
> | VG-LLM | ✘ | ✔ | 53.5 | 47.5 |
> | Omni-View | ✘ | ✘ | 50.8 | 45.0 |
>
> From this comparison, three observations can be made:
>
> - Omni-View outperforms methods that do not use any 3D-aware inputs (e.g., SPAR).
> - Omni-View's performance is slightly lower than methods that incorporate 3D-informed features such as VGGT (e.g., VG-LLM), suggesting that access to VGGT priors can provide a meaningful advantage.
> - Omni-View lags significantly behind methods that leverage explicit 3D inputs (e.g., Video-3D-LLM or Ross3D), which appear to offer the strongest benefit for 3D grounding.
>
> These results suggest that incorporating 3D-aware of 3D-informed signals generally improves 3D grounding performance. This raises the question: can Omni-View’s 3D visual grounding capability be further enhanced by introducing 3D-aware of 3D-informed inputs?
>
> To investigate, we conduct ablation studies by augmenting Omni-View with additional 3D-aware inputs following Video3DLLM or 3D-informed inputs following VG-LLM, then evaluate on the 3D scene understanding task. Results are as follows:
>
> | Methods | 3D input | VGGT input | Acc @ 0.25 | Acc @ 0.5 |
> | -- | -- | -- | -- | -- |
> | Omni-View | ✘ | ✘ | 50.8 | 45.0 |
> | Omni-View | ✘ | ✔ | 53.9 | 48.2 |
> | Omni-View | ✔ | ✘ | 58.2 | 50.4 |
>
> - When VGGT features are added into the input, Omni-View shows a moderate improvement in 3D visual grounding performance.
> - When camera poses are treated as position embedding following Video 3D-LLM, we observe a more significant performance gain.
>
> These findings confirm that integrating explicit 3D information benefits Omni-View in 3D visual grounding, consistent with trends observed in prior work.
>
> ---
>
> [1] Context as memory: Scene-consistent interactive long video generation with memory retrieval. Siggraph 2025.
>
> [2] Self Forcing: Bridging the Train-Test Gap in Autoregressive Video Diffusion. NeurIPS 2025.
>
> [3] Flow-GRPO: Training Flow Matching Models via Online RL. NeurIPS 2025.

---

> ### Author Response · Authors · 2025-11-20
>
> > Question 2. How well does Omni-View generalize to unseen or real-world multi-view scenes.
>
> **Answer.** Thank you for your thoughtful feedback. We would like to clarify that our dataset is primarily collected from ScanNet and RealEstate10K, *both of which consist of real-world scenes*. Following your valuable suggestion, we further evaluated the robustness of Omni-View on previously *unseen scenes*, assessing its performance in 3D visual reasoning and novel view synthesis.
>
> **1. 3D visual reasoning. ScanNet -> Replica.** Following the evaluation protocol of 3D-CLR [1], we tested Omni-View on the Replica dataset, which is a multi-view indoor scene dataset rendered by Habitat, rather than captured in the real world. Omni-View didn't see these synthetic images during training. We evaluate its ability to recognize objects (concept), count instances (counting), and reason about the spatial relationships between objects (relation and comparison). The results are shown in the following table.
>
> | Method | Concept | Counting | Relation | Comparison |
> |--|--|--|--|--|
> | NS-VQA | 58.6 | 19.2 | 29.7 | 58.1 |
> | 3D-CLR | 65.3 | 45.1 | 53.6 | 73.5 |
> | Omni-View (Ours)| 60.4 | 53.2 | 54.0 | 77.9 |
>
> From these results, we observe the following points.
>
> - *On object recognition (Concept)*, Omni-View underperforms compared to previous works. This is due to the presence of object categories in Replica that do not appear in training dataset we used (ScanNet), indicating limitations in generalization to unseen object types.
> - *On object counting (Counting)*, Omni-View achieves a significant improvement over prior methods, suggesting strong capabilities in enumerating objects even in unseen scenes.
> - *For reasoning about spatial relationships (Relation and Comparison)*, Omni-View outperforms existing approaches, demonstrating robust 3D spatial reasoning. This suggests that, despite potential challenges in recognizing certain unseen object categories, Omni-View maintains accurate geometric and relational reasoning for familiar objects within unseen scenes.
>
> In summary, while Omni-View may exhibit reduced generalization when encountering previously unseen object categories, it shows strong robustness in modeling spatial relationships among known objects, even in entirely unseen environments. This highlights the effectiveness of our Omni-View in 3D scene understanding.
>
> **2. Novel view synthesis. Indoor -> Outdoor.** Although all data in the RealEstate10K dataset are collected from the real world, they primarily consist of indoor scenes. To further evaluate Omni-View’s novel view synthesis capability in outdoor scenes, we conduct additional evaluation on ACID, a *real-world outdoor scene* dataset. The results are as follows:
>
> | Methods | PSNR | SSIM | LPIPS |
> | -- | -- | -- | -- |
> | ViewCrafter | 20.88 | 0.697 | 0.224 |
> | Voyager | 22.76 | 0.761 | 0.183 |
> | Omni-View (Ours) | 22.60 | 0.734 | 0.195 |
>
> From these results, we observe the following points.
>
> - *Quantitative results.* Omni-View demonstrates better generalization to outdoor scenes compared to ViewCrafter. However, its performance on outdoor scenes remains below that of Voyager. Given that Voyager is trained on a large amount of proprietary data including outdoor scenes, this performance gap is expected and understandable.
> - *Qualitative results.* In the revised Appendix A.4, we provide novel view synthesis results of Omni-View on outdoor scenes. When camera motion is small, Omni-View is able to generate novel views with consistent spatial and visual coherence.
>
> Experimental results show that although Omni-View was primarily trained on indoor scenes, it still exhibits generalization capability when applied to outdoor scenes.
>
> ---
>
> [1] 3D Concept Learning and Reasoning from Multi-View Images. CVPR 2023.

---

> > ### Comment · Reviewer_U2JG · 2025-11-25
> >
> > I appreciate the authors’ effort in preparing the rebuttal. The replies are so detailed. Most of my concerns and questions have been adequately addressed. I will increase my score to 6

---

> > > ### Author Response · Authors · 2025-11-26
> > >
> > > Thank you for your insightful comments and appreciation of our work and rebuttal. We will do our best to improve the final version of our paper based on your valuable suggestions.

---

### Official Review · Reviewer_Lkur · 2025-11-01

**Soundness:** 3
**Presentation:** 3
**Contribution:** 3
**Rating:** 8
**Confidence:** 4

**Summary:**

- This paper proposes Omni-View, a unified multimodal model that jointly performs 3D scene understanding, geometry estimation, and novel view synthesis from multiview images, based on the principle that generation facilitates understanding.

- This paper introduces a dual-path architecture consisting of a texture module (appearance generation) and a geometry module (depth/pose estimation), enabling bidirectional synergy between generative and understanding tasks.

- This paper uses a two-stage training strategy where joint training enhances understanding via generative signals, followed by refinement for high-quality 3D scene generation; achieves state-of-the-art performance on VSI-Bench and strong results in NVS and 3D Q&A.

**Strengths:**

- This paper demonstrates that generative 3D tasks (novel view synthesis, geometry estimation) can actively enhance 3D scene understanding, rather than being separate objectives.

- This paper has a unified architecture for 3D reasoning, with separate texture and geometry modules allow complementary learning of appearance and spatial structure, leading to better localization, spatial reasoning, and depth-aware Q&A.

- This paper outperforms specialized models in 3D understanding benchmarks while maintaining competitive NVS and scene generation performance, closing the gap between multimodal understanding and 3D generative models.

- A systematically organized evaluation and ablation study would strengthen the credibility of this paper.

**Weaknesses:**

- It would be beneficial to include a diagram that more precisely illustrates the functionality of each module and the architecture, compared to the current version.

- Additionally, visualizing 3D scene understanding / spatial reasoning / NVS from a single view as a video could also be an effective way to present the capabilities of the system.

- Is there a reason why you refer to Texture Module and Geometry Module in the equations (e.g., (eq. 1), (eq. 2)) without using italics? Also, I believe writing them as TextureModule and GeometryModule (without a space) would improve readability and be more suitable for mathematical notation.

**Questions:**

Mentioned in the weaknesses

---

> ### Author Response · Authors · 2025-11-21
>
> **Thank you for your recognition of Omni-View!** We respond to each of your comments one by one.
>
> > Weakness 1. It would be beneficial to include a diagram that more precisely illustrates the functionality of each module and the architecture, compared to the current version.
>
> **Answer.** Thank you for your valuable suggestion. **We have included a detailed description of Omni-View's technical design in Appendix B**, which outlines the input, output, and specific architectural configuration of each module.
>
> In addition, we provide a comprehensive textual explanation of the functionality of every module to ensure clarity and accessibility for the reader. Regarding the individual components:
>
> - **Understanding model.** It takes as input multi-view images of the current scene along with a textual question and autoregressively generates the corresponding textual answer.
> - **Texture module.** It receives a single image of the scene, a textual description, and a specified camera trajectory. It then autoregressively generates a video of the scene from the viewpoints defined by the trajectory, effectively imagining how the scene would appear from different angles.
> - **Geometry module.** It predicts depth maps and camera parameters for the predicted images of Texture module, using the latent denoised by the Texture module as contextual information. Since camera intrinsic and extrinsic parameters can be represented by a fixed-length token sequence (of length 9), we employ a learnable query mechanism to estimate camera parameters. Notably, during Stage 1 of training, we utilize features from the Understanding model corresponding to the input image as cross-attention keys to assist in this process.
>
> We believe these additions provide a more complete and transparent account of our model’s architecture and operation.

---

> ### Author Response · Authors · 2025-11-21
>
> > Weakness 2. Additionally, visualizing 3D scene understanding / spatial reasoning / NVS from a single view as a video could also be an effective way to present the capabilities of the system.
>
> **Answer.** Thank you for your valuable suggestion. **We have added visualization results in the Appendix A and supplementary materials** to demonstrate Omni-View's capabilities in 3D scene understanding, spatial reasoning, and novel view synthesis from a single view.
>
> - **3D scene understanding.** We provide visual examples in Appendix A.1, including tasks such as scene question answering and 3D object detection. For the detection task, we visualize the predicted 3D bounding boxes to demonstrate the accuracy of the estimated coordinates.
> - **Spatial reasoning.** We present qualitative results in Appendices A.2 and A.3, covering tasks in VSI-Bench. Omni-View supports multiple output formats, including free-form text, numerical values, and multiple-choice responses. Additionally, in Appendix A.3, we include an analysis of scene-consistency in absolute scale depth prediction task, showing that Omni-View is capable of estimating absolute depth even across scenes with different metric scales.
> - **NVS from a single view.** We present results in Appendices A.4 and the supplementary materials. Appendix A.4 showcases Omni-View’s performance in both indoor and outdoor scenes, along with several failure cases where significant camera motion poses challenges. The supplementary materials include the videos synthesized by Omni-View, all of them have been anonymized to protect privacy.
>
> We hope these additions provide a clearer and more comprehensive picture of Omni-View's capabilities.

---

> ### Author Response · Authors · 2025-11-21
>
> > Weakness 3. Is there a reason why you refer to Texture Module and Geometry Module in the equations (e.g., (eq. 1), (eq. 2)) without using italics? Also, I believe writing them as TextureModule and GeometryModule (without a space) would improve readability and be more suitable for mathematical notation.
>
> **Answer.** We originally used plain text without italics to indicate that these terms refer to functional components of the model rather than mathematical variables or operators. However, we agree that this choice may introduce inconsistency in readability and could be improved for better integration into mathematical notation.
>
> Following your suggestion, in the revised manuscript, we have unified the notation by renaming these terms as TextureModule and GeometryModule (without spaces) wherever they appear in equations. **Our revised PDF has been submitted.** Please see the Eq.1 and Eq.2 in our latest submission (PDF file) for more details.

---

> > ### Comment · Reviewer_Lkur · 2025-11-26
> >
> > Thanks for your comment. I will maintain my score.

---

> > > ### Author Response · Authors · 2025-11-26
> > >
> > > Thank you again for your time and insightful feedback. We will revise our manuscript carefully based on your valuable comments.

---

### Official Review · Reviewer_Meww · 2025-11-02

**Soundness:** 3
**Presentation:** 3
**Contribution:** 3
**Rating:** 6
**Confidence:** 4

**Summary:**

The paper presents Omni-View, a unified model for 3D scene understanding and generation from multiview images that tests the “generation facilitates understanding” hypothesis. Built on Bagel, it splits generation into a texture module (flow matching with Plücker pose encoding, autoregressive NVS) and a geometry module (depth and camera pose via flow matching with cross-attention to understanding features). A two-stage training recipe first jointly trains understanding/texture/geometry with a dense-to-sparse curriculum, then fine-tunes generation with RGB-Depth-Pose joint learning. Omni-View achieves SOTA on VSI-Bench (55.4), improves QA/localization versus unified baselines without 3D inputs, and delivers strong NVS/scene generation results on Re10k.

**Strengths:**

- The paper presents a unified 3D understanding–generation framework that cleanly separates texture and geometry, a simple yet original design that operationalizes “generation facilitates understanding.”
- The two-stage recipe with dense-to-sparse curriculum and autoregressive NVS is well-motivated, technically sound, and shows careful loss design and gradient routing to benefit the understanding model.
- Writing is clear and structured, with concrete training details, datasets, metrics, and ablations that isolate the contribution of each module and training choice.
- The empirical significance is strong, with SOTA results on VSI-Bench and competitive 3D QA/localization, plus solid NVS and scene generation metrics, demonstrating broad impact across 3D reasoning and generation.

**Weaknesses:**

- Limited novelty relative to prior unified frameworks (Bagel, VILA-U, BLIP3o, Harmon)
The core idea of leveraging generation to aid understanding has precedents in 2D unified models and recent 3D works that inject reconstruction priors (e.g., Ross3D; VG-LLM/Spatial-MLLM via VGGT features). The split into texture vs. geometry resembles established “appearance vs. structure” decouplings in 3D pipelines (e.g., ViewCrafter, Voyager). Clarify what is fundamentally new beyond integrating these pieces within Bagel, and compare to a “single-branch with multi-heads” backbone.

- Ambiguity in camera control and absolute metric grounding
The paper reports strong perceptual metrics but acknowledges difficulty in precise camera control and absolute depth scale. Because the gains on VSI categories like Abs. Dist. hinge on metric grounding, add analyses: scale consistency across scenes, depth-scale calibration via known baselines, and camera-pose accuracy vs. ground truth under diverse motions.

- Dataset overlap and generalization concerns
Though the authors state they avoid using understanding images for generation training, several datasets share scene domains with Re10k-like indoor content, risking leakage of priors. Please report cross-dataset generalization (e.g., ScanNet -> Replica, RealEstate10K -> ACID/CO3D subsets) to support robustness claims.

- Incomplete ablations on design choices and routing
The geometry module conditions only on the last-layer texture latent and uses cross-attention to the understanding model. Test alternatives: multi-scale latents, earlier-layer features, and gating that controls gradient flow to avoid potential interference. Provide compute/latency breakdowns for stage 1 vs. stage 2, and show sensitivity to $\lambda_{geo}$, pose-query design, and Plücker vs. other pose encodings.

**Questions:**

- Clarify the novelty beyond architectural decoupling: In what ways is the texture/geometry split more than a clean engineering separation compared to prior “appearance vs. structure” decouplings (e.g., ViewCrafter, Voyager) and unified frameworks (BAGEL, VILA-U, BLIP3o, Harmon)? Could you provide a controlled comparison to a single-branch generator with two prediction heads (texture, geometry) at equal parameter count?
- The ablations show AR improves spatiotemporal reasoning. Can you report exposure-bias analyses at inference time, e.g., teacher-forcing vs. free-running rollouts? Does diffusion forcing mitigate compounding errors, and how does performance vary with rollout length (8/16/32 frames)? Have you tested scheduled sampling or token-level AR only on camera poses while keeping texture bidirectional?
- Provide qualitative and quantitative failure analyses: cases where geometry improves understanding but harms texture fidelity (and vice versa), per-category VSI error tied to pose/depth errors, and sensitivity to large viewpoint changes where you noted inconsistencies. Would integrating a small, explicit 3D proxy (e.g., Gaussians or sparse point clouds) at training but not inference close these gaps?

---

> ### Author Response · Authors · 2025-11-20
>
> Thanks for your insightful comments. We list your advice and questions followed by detailed answers, then **argue that your concerns can be addressed**.
>
> > Weakness 1 & Question 1. Limited novelty relative to prior unified frameworks (Bagel, VILA-U, BLIP3o, Harmon). The core idea of leveraging generation to aid understanding has precedents in 2D unified models and recent 3D works that inject reconstruction priors (e.g., Ross3D; VG-LLM/Spatial-MLLM via VGGT features). The split into texture vs. geometry resembles established “appearance vs. structure” decouplings in 3D pipelines (e.g., ViewCrafter, Voyager). Clarify what is fundamentally new beyond integrating these pieces within Bagel, and compare to a “single-branch with multi-heads” backbone. Clarify the novelty beyond architectural decoupling: In what ways is the texture/geometry split more than a clean engineering separation compared to prior “appearance vs. structure” decouplings (e.g., ViewCrafter, Voyager) and unified frameworks (BAGEL, VILA-U, BLIP3o, Harmon)? Could you provide a controlled comparison to a single-branch generator with two prediction heads (texture, geometry) at equal parameter count?
>
> **Answer.** We want to clarify the novelty of Omni-View relative to prior studies in four sections. In the first three sections, we compare Omni-View with previous 3D scene understanding methods (Ross3D, VG-LLM, and Spatial-MLLM), previous 3D scene generation methods (ViewCrafter and Voyager), and previous unified frameworks on 2D vision to demonstrate the novelty of our method in terms of paradigm and technical details. Finally, we provide a controlled comparison to a single-branch generator with two prediction heads (texture and geometry) at equal parameter count.
>
> **(1/4) We argue that Omni-View differs from previous 3D scene understanding methods in paradigm.** We would like to clarify that previous works (Ross3d, VG-LLM, Spatial-MLLM) explore "reconstruction facilitates understanding", while our Omni-View demonstrates "generative facilitates understanding".
>
> We start from distinguishing between generation and reconstruction.
>
> - *Generation.* Creating or imagining new content. It requires diverse outputs that conform to semantic or physical laws.
> - *Reconstruction.* Restoring the original input signal or its characteristics as much as possible. It requires outputs with high fidelity.
>
> Based on this strict defination, let's re-think our Omni-View and previous works (Ross3d, VG-LLM, Spatial-MLLM):
>
> 1. Ross3D explores "*reconstructive objective*", as emphasized in Ross3D's title ("Reconstructive Visual Instruction Tuning with 3D-Awareness"). Besides the understanding task, Ross3D's training objective is to recover the signals of the masked viewpoints or the BEV of the current scene from a lots of known viewpoints. This conflicts with the nature of the generation (creating or imagining new content). Furthermore, in Ross, the predecessor to Ross3D, they also emphasized the difference between Ross and generative tasks in the Figure 11 of its Appendix. Going further, we also compared Ross with Ross3D's reconstructive objective in our Table 6. The results demonstrate that the generative objective is more effective in 3D scenes.
> 2. Methods using VGGT (VG-LLM / Spatial-MLLM) explore "*reconstruction feature addition*". First, these methods only add 3D features at the input without additional training objectives, which is significantly different from Omni-View and Ross3D. Second, VGGT is a powerful 3D reconstruction model, which cannot be considered a "generation model". As stated on page 2, lines 7-9 of the original paper of VGGT: "Visual Geometry Grounded Transformer (VGGT), a feed-forward neural network that performs 3D reconstruction."
> 3. **In contrast**, our Omni-View explores "*generative objectives*". Besides the understanding task, our training objective is as follows: starting from a single view image, to *generate* reasonable images and their geometries after the viewpoint transformation.
>
> | Methods | additional generative objective | additional reconstructive objective | additional VGGT input |
> | -- | -- | -- | -- |
> | Ross3D | ✘ | ✔ | ✘ |
> | VG-LLM / Spatial-MLLM | ✘ | ✘ | ✔ |
> | Omni-View (Ours) | ✔ | ✘ | ✘ |
>
> **In summary**, as shown in the above table, Omni-View differs significantly from previous methods in its paradigm, primarily in that we use generative objectives, while other methods use reconstructive objectives or directly use features from the reconstruction model as input without adding new training objectives.

---

> ### Author Response · Authors · 2025-11-20
>
> > Weakness 1 & Question 1.
>
> **(2/4) We argue that Omni-View differs from previous 3D scene generation methods in technical details on decoupling appearance and structure.**
>
> 1. Viewcrafter also uses separate modules to model appearance and structure. It uses a pre-trained depth estimation network to estimate the depth of a reference viewpoint, unprojects it into a point cloud, uses this point cloud to render the target viewpoint, and finally performs inpainting on the rendered image. However, due to the non-differentiable nature of point cloud rendering and unprojection, gradients cannot propagate between the depth estimation network and the image inpainting network, *preventing Viewcrafter from jointly optimizing these two modules*.
> 2. *Voyager does not decouple appearance and structure*. Voyager stitches the image and its depth map together in height and uses a video diffusion model to model the stitched image. Voyager uses one model and one output head to process both the appearance and structure.
> 3. **In contrast**, Omni-View uses separate modules to model appearance and structure. The Texture module generates RGB information, while the Geometry module predicts depth information and camera pose. These two modules are jointly optimized during training.
>
> | Methods | decoupling appearance and structure | separate modules | jointly optimization |
> | -- | -- | -- | -- |
> | ViewCrafter | ✔ | ✔ | ✘ |
> | Voyager | ✘ | ✘ | ✔ |
> | Omni-View (Ours) | ✔ | ✔ | ✔ |
>
> The table above more clearly illustrates the differences between our Omni-View, ViewCrafter, and Voyager.
>
> ---
>
> **(3/4) We argue that prior unified models (BAGEL, VILA-U, BLIP3o, Harmon) focus on coexisting understanding and generation, while Omni-View focuses on "generation facilitates understanding" based on them.**
>
> - Previous unified models, such as Bagel, VILA-U, BLIP3o, and Harmon, aim to coexist understanding and generation. Specifically, VILA-U and Harmon focus on integrating the two tasks at the image tokenizer level, while Bagel and BLIP3o explore architectural designs in the backbone to support both functionalities. *However, none of these approaches explicitly examine whether or how generative training objectives can actively enhance visual understanding performance.*
> - More recently, some studies (e.g., UAE [1]) have begun to explore the idea of "generation facilitates understanding" in 2D vision. In these works, SigLIP features of an image are fed into the MLLM, and the MLLM is trained to reconstruct either pixel values or VAE latents of the original image. This essentially amounts to recover the original signal, rather than generating novel or imagined content. Therefore, strictly speaking, *these methods do not embody true generative training objective.*
> - **In contrast**, Omni-View makes a distinct and novel contribution: *we explicitly propose and demonstrate that incorporating generative training objective can effectively improve the model’s spatial understanding.* To the best of our knowledge, this insight has neither been clearly articulated nor empirically validated in prior unified models. Our work highlights a new direction where novel view generation serves as a supervisory signal for enhancing scene understanding, going beyond "coexistence" patten. **We have provided a more comprehensive analysis of this comparison** in the first three paragraphs of the Introduction.
>
> ---
>
> [1] Can Understanding and Generation Truly Benefit Together -- or Just Coexist?

---

> ### Author Response · Authors · 2025-11-20
>
> > Weakness 1 & Question 1.
>
> **(4/4) Omni-View works better than single-branch with multi-heads at equal parameter count.**
>
> We would like to first clarify that the comparison with the "single-branch with multi-heads" approach has already been included in the gray row of Table 4. As shown there, Omni-View achieves better performance on 3D scene understanding tasks compared to this baseline.
>
> However, we fully agree with the reviewer’s observation that our experiments in Table 4 were not conducted under equal parameter settings. Specifically, Omni-View introduces an additional 1B-parameter geometry module during training, whereas the "single-branch with multi-heads" method only adds a lightweight prediction head. This discrepancy in model capacity could potentially introduce bias in favor of our approach.
>
> To address this issue, we have attempted to create a fairer comparison by scaling up the parameters in the "single-branch with multi-heads" baseline, aiming to match the total number of trainable parameters during training. Since Omni-View is built upon a fixed pre-trained backbone, directly scaling the backbone is not feasible. Instead, we choose to *scale up the geometry head* in the baseline to better align the overall parameter count during training. The updated experimental results are provided in the revised manuscript.
>
> | Method | Params | SQA3D | ScanQA | ScanRefer | VSI-Bench (subset) |
> | -- | -- | -- | -- | -- | -- |
> | Metrics / Subtasks |  | EM | CIDEr / BLEU-4 | Acc @ 0.25 | Obj. Cnt. / Abs. Dist. / Obj. Size / Rel. Dist. / Appr. Order |
> | Omni-View | 8B | 59.2 | 103.0 / 16.2 | 50.8 | 70.3 / 46.4 / 68.6 / 65.9 / 49.0 |
> | single-branch with multi-heads | 7B | 58.7 | 99.2 / 15.0 | 49.0 | 69.5 / 45.9 / 67.8 / 63.8 / 48.2 |
> | single-branch with multi-heads (larger heads) | 8B | 58.8 | 99.6 / 15.4 | 49.6 | 69.8 / 46.0 / 67.7 / 64.2 / 47.9 |
>
> The results demonstrate that **Omni-View outperforms the "single-branch with multi-heads" paradigm under comparable training-time parameter counts**.

---

> ### Author Response · Authors · 2025-11-20
>
> > Weakness 2. Ambiguity in camera control and absolute metric grounding. The paper reports strong perceptual metrics but acknowledges difficulty in precise camera control and absolute depth scale. Because the gains on VSI categories like Abs. Dist. hinge on metric grounding, add analyses: scale consistency across scenes, depth-scale calibration via known baselines, and camera-pose accuracy vs. ground truth under diverse motions.
>
> **Answer.** After careful analysis of your comments, we think there may be a misunderstanding regarding the output format of the VSI-Bench. We would like to clarify that, in the official evaluation protocol of VSI-Bench, the predicted absolute distance is expressed in texts, e.g., a textual number, rather than as pixel-level depth maps. We provide visual illustrations in the Appendix A.3 to further clarify this output format.
>
> Based on this clarification, we analyze the following:
>
> - **Depth-scale calibration via known baselines.** Because the output is text rather than pixel level depth map, the evaluation *does not require depth-scale calibration* to recover metric-aligned absolute scales from relative depth predictions. The model directly outputs human-interpretable estimates of size or distance, which are evaluated against ground truth values using semantic and numerical matching rules provided by VSI-Bench.
> - **Scale consistency across scenes.** However, we can still analyze scale consistency across different scenes. To this end, we selected 200 samples from the SPAR-7M dataset and used Omni-View to perform absolute distance prediction based on multi-view images. This test set consists of data from three scene datasets: ScanNet and ScanNet++, where the average scene depth ranges from 3 to 5 meters with a maximum depth of 12 meters; and Structured3D, which features larger-scale environments with average depths between 4 and 6 meters and maximum depths reaching up to 20 meters.
>
> | Method | Depth-OC-MV in SPAR (scannet & scannetpp) | Depth-OC-MV in SPAR (structured3d) |
> | -- | -- | -- |
> | Omni-View | 65.2 | 58.0 |
>
> As shown, Omni-View exhibits a moderate performance drop on the Structured3D portion of SPAR, but the decline is not significant. This suggests that the model maintains reasonable generalization in absolute depth estimation across scenes with varying depth ranges.
>
> In addition, we provide qualitative visualizations demonstrating that Omni-View can produce reasonably accurate absolute scale predictions across datasets with different metric scales, such as ScanNet and Structured3D. These examples can be found in the revised Appendix A.3.
>
> - **Camera-pose accuracy vs. ground truth under diverse motions.** We evaluate the camera pose accuracy of Omni-View on video datasets with accurate camera poses: Re10k and Tanks & Temples. Among these, Tanks & Temples consist of outdoor scenes, with Tanks & Temples featuring larger camera motions. We evaluate them using the standard metric AUC@30, which combines RRA and RTA. RRA (Relative Rotation Accuracy) and RTA (Relative Translation Accuracy) calculate the relative angular errors in rotation and
> translation, respectively.
>
> | Dataset | Re10k | Tanks & Temples |
> | -- | -- | -- |
> | VGGT | 85.3 | 87.8 |
> | Omni-View | 85.6 | 82.5 |
>
> As shown in the results, Omni-View demonstrates reasonable generalization to outdoor scenes and scenes with diverse camera motions, despite being primarily trained on indoor scenes. However, its camera pose estimation accuracy is weaker compared to 3D reconstruction model, like VGGT. This performance gap is expected, as the motivation for designing the geometry module was to investigate whether jointly learning a geometry estimation task could enhance scene understanding performance, rather than to achieve state-of-the-art accuracy on the geometry estimation task itself.
>
> ---
>
> **In the end**, we believe these questions reflect your deeper concern: ***whether the model has overfitted to absolute scale information specific to the VSI scenes (ScanNet).*** We would like to clarify that, due to *the inclusion of more general video data such as llava-hound in our training set*, Omni-View **does not specifically fit** to the characteristics of datasets like ScanNet. The detailed composition of our training data is provided in our [response to reviewer U2JG Weakness 2](https://openreview.net/forum?id=pDu6u9cnEB&noteId=RsznraZYg8).

---

> ### Author Response · Authors · 2025-11-20
>
> > Weakness 3. Dataset overlap and generalization concerns. robustness claim. Though the authors state they avoid using understanding images for generation training, several datasets share scene domains with Re10k-like indoor content, risking leakage of priors. Please report cross-dataset generalization (e.g., ScanNet -> Replica, RealEstate10K -> ACID/CO3D subsets) to support robustness claims.
>
> **Answer.** Thank you for your kind advice. Based on your valuable suggestions, we validated the robustness of Omni-View in 3D visual reasoning and novel view synthesis.
>
> **1. 3D visual reasoning. ScanNet -> Replica.** We tested Omni-View on the Replica dataset, which is a multi-view indoor scene dataset rendered by Habitat. We evaluate its ability to recognize objects (concept), count objects (counting), and reason about the spatial relationships between objects (relation and comparison) following 3D-CLR [1]. The results are shown in the following table.
>
> | Method | Concept | Counting | Relation | Comparison |
> |--|--|--|--|--|
> | NS-VQA | 58.6 | 19.2 | 29.7 | 58.1 |
> | 3D-CLR | 65.3 | 45.1 | 53.6 | 73.5 |
> | Omni-View (Ours)| 60.4 | 53.2 | 54.0 | 77.9 |
>
> From these results, we observe the following points.
>
> - *On object recognition (Concept)*, Omni-View underperforms compared to previous works. This is due to the presence of object categories in Replica that do not appear in training dataset we used (ScanNet), indicating limitations in generalization to unseen object types.
> - *On object counting (Counting)*, Omni-View achieves a significant improvement over prior methods, suggesting strong capabilities in enumerating objects even in unseen scenes.
> - *For reasoning about spatial relationships (Relation and Comparison)*, Omni-View outperforms existing approaches, demonstrating robust 3D spatial reasoning. This suggests that, despite potential challenges in recognizing certain unseen object categories, Omni-View maintains accurate geometric and relational reasoning for familiar objects within unseen scenes.
>
> **In summary**, while Omni-View may exhibit reduced generalization when encountering previously unseen object categories, it shows strong robustness in modeling spatial relationships among known objects, even in entirely unseen environments. This highlights the effectiveness of our Omni-View in 3D scene understanding.
>
> **2. Novel view synthesis. RealEstate10K -> ACID.** To further evaluate Omni-View’s novel view synthesis capability in outdoor scenes, we conduct additional evaluation on *ACID, a real-world outdoor scene* dataset. The results are as follows:
>
> | Methods | PSNR | SSIM | LPIPS |
> | -- | -- | -- | -- |
> | ViewCrafter | 20.88 | 0.697 | 0.224 |
> | Voyager | 23.06 | 0.771 | 0.173 |
> | Omni-View (Ours) | 22.60 | 0.734 | 0.195 |
>
> - *Quantitative results.* Omni-View demonstrates better generalization to outdoor scenes compared to ViewCrafter. However, its performance on outdoor scenes remains below that of Voyager. Given that Voyager is trained on a large amount of proprietary data including outdoor scenes, this performance gap is expected.
> - *Qualitative results.* In the revised Appendix A.4, we provide novel view synthesis results of Omni-View on outdoor scenes. When camera motion is small, Omni-View is able to generate novel views with consistent spatial and visual coherence.
>
> **Our revised PDF has been submitted.** Please see the Appendix A.4 in our latest submission (PDF file) for more details.
>
> ---
>
> [1] 3D Concept Learning and Reasoning from Multi-View Images. CVPR 2023.

---

> ### Author Response · Authors · 2025-11-20
>
> > Weakness 4. Incomplete ablations on design choices and routing. The geometry module conditions only on the last-layer texture latent and uses cross-attention to the understanding model. Test alternatives: multi-scale latents, earlier-layer features, and gating that controls gradient flow to avoid potential interference. Provide compute/latency breakdowns for stage 1 vs. stage 2, and show sensitivity to , pose-query design, and Plücker vs. other pose encodings.
>
> **Answer.** Thank you for raising these insightful design considerations. Due to the substantial computational cost of full-scale training, we were unable to complete all ablations on the entire dataset within the rebuttal period. *As a practical alternative, we randomly selected 20% of the training data*, which allows us to obtain preliminary yet indicative trends in model behavior while significantly reducing training time and resource requirements. Below, we present and analyze the results of the **six ablations you suggested** and the **computational cost for stage 1 and stage 2**.
>
> **(1/7) Multi-scale texture latents into geometry module.** Experimental results show that feeding multi-scale texture latents into the geometry module indeed further improves the model's performance on 3D scene understanding tasks.
>
> | multi-scale latents into geometry module | SQA3D (EM) | ScanQA (CIDEr) | ScanRefer (Acc @ 0.25) | VSI-Bench (avg.) |
> | -- | -- | -- | -- | -- |
> | ✘ (ours) | 57.4 | 93.4 | 42.3 | 44.1 |
> | ✔ | 58.0 | 96.0 | 44.0 | 44.4 |
>
> ---
>
> **(2/7) Early-layer texture latent into geometry module.** Experimental results show that using early-layer texture latents performs worse on 3D scene understanding tasks compared to directly using the final-layer texture latent features.
>
> | layer index | SQA3D (EM) | ScanQA (CIDEr) | ScanRefer (Acc @ 0.25) | VSI-Bench (avg.) |
> | -- | -- | -- | -- | -- |
> | last (ours) | 57.4 | 93.4 | 42.3 | 44.1 |
> | first | no convergence |
> | second last | 57.6 | 92.8 | 41.0 | 43.7 |
>
> ---
>
> **(3/7) Understanding latent injection: latent gating vs. cross attention.** We find that gating significantly impairs the model's understanding performance. We believe this is because, to prevent numerical overflow, the geometry module scales down the numerical values of the gate matrix from understanding model. This step slows down the optimization, ultimately leading to degraded model performance.
>
> | understanding latent injection | SQA3D (EM) | ScanQA (CIDEr) | ScanRefer (Acc @ 0.25) | VSI-Bench (avg.) |
> | -- | -- | -- | -- | -- |
> | cross attention (ours) | 57.4 | 93.4 | 42.3 | 44.1 |
> | latent gating | 53.0 | 88.9 | 38.4 | 36.5 |
>
> ---
>
> **(4/7) Sensitivity to $\lambda_{geo}$.** We conducted an ablation study on the coefficient of the geometry loss. The results show that the model achieves better performance when $\lambda_{geo} \le 0.1$.
>
> | $\lambda_{geo}$ | SQA3D (EM) | ScanQA (CIDEr) | ScanRefer (Acc @ 0.25) | VSI-Bench (avg.) |
> | -- | -- | -- | -- | -- |
> | 0.01 | 57.3 | 94.5 | 42.6 | 44.3 |
> | 0.1 (ours) | 57.4 | 93.4 | 42.3 | 44.1 |
> | 1 | 56.6 | 92.8 | 42.0 | 42.7 |
> | 10 | 53.8 | 91.1 | 39.7 | 40.4 |
>
> ---
>
> **(5/7) Sensitivity to pose-query design.** We clarify that we use a learnable query approach to model camera poses. This approach does not involve additional design choices or auxiliary mechanisms.
>
> ---
>
> **(6/7) Plücker vs. other camera pose encodings.** We have compared our approach with the naive camera pose encoding method proposed in CAT4D [1]. Experimental results show that the choice of camera pose encoding has little impact on performance for understanding tasks, but leads to a slight improvement in generation tasks.
>
> - *Understanding.*
> | camera pose encoding | SQA3D (EM) | ScanQA (CIDEr) | ScanRefer (Acc @ 0.25) | VSI-Bench (avg.) |
> | -- | -- | -- | -- | -- |
> | Plücker (ours) | 57.4 | 93.4 | 42.3 | 44.1 |
> | Naive | 57.4 | 93.3 | 42.5 | 44.1 |
>
> - *Generation.*
> | camera pose encoding | PSNR | SSIM | LPIPS |
> | -- | -- | -- | -- |
> | Plücker (ours) | 23.22 | 0.817 | 0.114 |
> | Naive | 23.24 | 0.820 | 0.112 |
>
> [1] CAT4D: Create Anything in 4D with Multi-View Video Diffusion Models. CVPR 2025.
>
> ---
>
> **(7/7). Computational cost.**
>
> We present the computational cost of Omni-View from both training and inference perspectives.
>
> **Training:**
> - *Stage 1:* We train on the 3D scene understanding dataset for 10,000 iterations with a packed sequence length of 100k, using 32 H100 GPUs, which takes approximately 160 hours.
> - *Stage 2:* For the generation task, we perform 20,000 iterations on the novel view synthesis dataset with a packed sequence length of 32k, using 32 H100 GPUs, requiring approximately 40 hours in total.
>
> **Inference:**
> - *Understanding:* The model takes approximately 2.5 seconds on average to process a 32-frame multi-view scene understanding query using a single H100 GPU.
> - *Generation:* Generating a single image at resolution 640 × 352 takes about 2.2 seconds on average with one H100 GPU.

---

> ### Author Response · Authors · 2025-11-22
>
> > Question 2. The ablations show AR improves spatiotemporal reasoning. Can you report exposure-bias analyses at inference time, e.g., teacher-forcing vs. free-running rollouts? Does diffusion forcing mitigate compounding errors, and how does performance vary with rollout length (8/16/32 frames)? Have you tested scheduled sampling or token-level AR only on camera poses while keeping texture bidirectional?
>
> **Answer.** We analyze the potential exposure bias in Omni-View's inference process by examining both understanding and generation tasks.
>
> **(1/3) Exposure bias in spatiotemporal reasoning.**
>
> We would like to clarify that when Omni-View performs inference on tasks requiring spatiotemporal reasoning, e.g. the appearance order task in VSI-Bench, it does not require autoregressive novel view synthesis. *As a result, exposure bias does not arise in this setting.*
>
> ---
>
> **(2/3) Exposure bias in novel view synthesis.**
>
> While our experimental results (e.g., Table 5 in the main text) demonstrate that adding autoregressive novel view synthesis (NVS) training objective can enhance the model's spatiotemporal reasoning capabilities, we fully acknowledge that it introduces accumulated errors during NVS inference. This issue arises from the *inherent inconsistency between training and inference in autoregressive generation.* During training, the model uses teacher forcing and observes ground-truth previous frames. In contrast, during inference, it operates in a free-running manner, relying entirely on its own generated frames, which may contain errors from prior steps.
>
> To mitigate this problem, Omni-View adopts diffusion forcing, a technique that has been widely shown to *effectively mitigate, though not completely eliminate, accumulated errors* in autoregressive video generation. To better understand its impact on generation quality, we conducted additional **evaluation on NVS under different rollout lengths**: 8, 16, 25, and 32 frames. The results are summarized below.
>
> | Rollout lengths | PSNR | SSIM | LPIPS |
> | -- | -- | -- | -- |
> | 8 | 24.51 | 0.833 | 0.105 |
> | 16 | 24.27 | 0.821 | 0.107 |
> | 25 (default setting in Omni-View) | 23.22 | 0.817 | 0.114 |
> | 32 | 22.39 | 0.797 | 0.119 |
>
> As shown, performance remains relatively stable up to 16 frames, with only marginal degradation in metrics. This indicates that visual consistency is preserved and accumulated error is not yet pronounced. However, when extending to 25 and 32 frames, we observe a clear decline in PSNR, SSIM and LPIPS, suggesting that errors accumulate over frame and increasingly affect the quality of generated frames.
>
> **However**, we have explored approaches to mitigate these accumulated errors. Specifically, we introduced a "generation in grid" strategy to improve temporal consistency and reduce error propagation. For a detailed explanation and visual illustrations, please refer to [our response to Reviewer U2JG’s comment on Weakness 3-2](https://openreview.net/forum?id=pDu6u9cnEB&noteId=QjIip97JAC) and the revised Appendix A.5.
>
> ---
>
> **(3/3) Scheduled sampling strategies in NVS.**
>
> Following your suggestion, **we explored various scheduled sampling strategies to try to alleviate exposure bias in NVS**. Since our generation is based on flow matching, scheduled sampling mainly involves adjusting the timeshift hyperparameter and experimenting with different schedulers. We are currently evaluating these alternatives on 25 rollout lengths and the results are shown below.
>
> - *Timeshift.*
> | timeshift | PSNR | SSIM | LPIPS |
> | -- | -- | -- | -- |
> | 1.0 | 23.30 | 0.818 | 0.116 |
> | 1.5 (ours) | 23.22 | 0.817 | 0.114 |
> | 2.0 | 23.15 | 0.815 | 0.113 |
>
> - *Scheduler.*
> | scheduler | PSNR | SSIM | LPIPS |
> | -- | -- | -- | -- |
> | FlowMatchEulerDiscreteScheduler (ours) | 23.22 | 0.817 | 0.114 |
> | FlowMatchHeunDiscreteScheduler [1] | 23.24 | 0.816 | 0.115 |
>
> The above experimental results indicate that scheduled sampling has limited impact on the NVS performance.
>
> ---
>
> **(4/4) Token-level AR.**
>
> We argue that adopting token-level autoregression would require replacing the pre-trained backbone model, which entails significant engineering efforts and training costs. Due to these practical constraints, we were unable to complete this experiment during the rebuttal period. As a result, we are currently unable to verify the actual effectiveness of token-level autoregression applied only to camera poses, or its precise impact on exposure bias. We acknowledge this as an important direction for future work and plan to investigate it thoroughly in future works.
>
> ---
>
> [1] Scaling rectified flow transformers for high-resolution image synthesis. ICML 2024.

---

> ### Author Response · Authors · 2025-11-22
>
> > Question3. Provide qualitative and quantitative failure analyses: cases where geometry improves understanding but harms texture fidelity (and vice versa), per-category VSI error tied to pose/depth errors, and sensitivity to large viewpoint changes where you noted inconsistencies. Would integrating a small, explicit 3D proxy (e.g., Gaussians or sparse point clouds) at training but not inference close these gaps?
>
> **Answer.** We would like to answer your questions one by one.
>
> **(1/3) Failure cases where geometry improves understanding but harms texture fidelity (and vice versa) and per-category VSI error tied to pose/depth errors.**
>
> We clarify that such failure cases do not occur in our framework. This is because the understanding model and the geometry module are decoupled during inference, meaning that neither relies on the other’s outputs as input or intermediate representations. For example, when evaluated on VSI-Bench tasks such as object size estimation, the understanding model directly generates textual responses without accessing poses or pixel-level depth maps produced by the geometry module. As a result, errors or artifacts in the geometry module do not propagate to the understanding model during inference.
>
> Similarly, the texture module and the geometry module are also decoupled during inference. This means that improvements in geometric reasoning do not compromise texture fidelity, nor does enhanced texture generation degrade geometric accuracy. Each module operates independently at inference time, ensuring clean separation of responsibilities and avoiding undesirable trade-offs.
>
> We emphasize that **the performance improvement is achieved by introducing additional training objectives, not by providing extra inputs to the model**. This design ensures that the enhanced capabilities stem from better supervision during training, rather than from architectural dependencies or privileged information at inference time.
>
> ---
>
> **(2/3) Sensitivity to large viewpoint changes.**
>
> We analyzed the sensitivity of generation performance to large viewpoint changes by examining how model outputs vary with the degree of camera motion. Specifically, we measure performance as a function of field-of-view (FOV) overlap [1] between input and target views, and report the corresponding PSNR, SSIM, and LPIPS.
>
> - *Quantitative results.*
> The results reveals that as viewpoint differences increase (smaller FOV overlap), the quality of novel view synthesis degrades noticeably. This indicates that the model faces greater challenges when synthesizing views under large camera motions.
>
> | FOV overlap | PSNR | SSIM | LPIPS |
> | -- | -- | -- | -- |
> | 0.4 $\sim$ 0.6 | 21.04 | 0.737 | 0.165 |
> | 0.6 $\sim$ 0.8 | 23.98 | 0.820 | 0.107 |
>
> - *Qualitative results.*
> In Appendix A.4, we present failure cases under extreme viewpoint changes. As illustrated in the figure in the revised Appendix A.4, the generated images under extreme camera motion exhibit visible artifacts, which are highlighted within red dashed boxes. These cases suggest that maintaining geometric coherence and texture consistency becomes increasingly difficult under wide-baseline settings.
>
> - *Future works.*
> Based on these results, we will focus on addressing two key challenges to improve performance on outdoor scenes with extensive camera motion in future works: (1) developing more precise camera control mechanisms to better handle large pose variations, and (2) enhancing inter-frame texture consistency to improve visual stability across distant views.
>
> ---
>
> **(3/3) Integrating a small, explicit 3D proxy at training but not inference.**
>
> We believe that introducing an explicit 3D proxy during training but not using it during inference *leads to a mismatch between training and inference*, which can result in significant accumulated errors. This discrepancy negatively affects model performance and is unlikely to effectively close the performance gaps.
>
> ---
>
> [1] Context as memory: Scene-consistent interactive long video generation with memory retrieval. Siggraph 2025.

---

> ### Public Comment · ~Baiqiao_Yin1 · 2025-11-24
>
> The replies are so detailed.

---

> ### Comment · Reviewer_Meww · 2025-11-25
>
> I give the authors a thumbs up for providing such a detailed response. I believe my concerns have been addressed and will raise my score accordingly.

---

> > ### Author Response · Authors · 2025-11-26
> >
> > Thank you for your thoughtful feedback and the thumbs up. We are pleased that our responses resolved your concerns, and we will integrate your valuable suggestions into the final version of the paper.

---

### Meta-Review · Area_Chair_tS1Z · 2025-12-28

**Summary:**

The paper proposes Omni-View, a unified framework for 3D scene understanding and generation. Reviewers praised the intuitive motivation  and the modular architecture. Initial concerns focused on the novelty relative to existing unified frameworks (Bagel), the lack of qualitative geometry visualizations (depth/pose), implementation transparency, and the mechanism behind the performance gains. The rebuttal was highly effective: authors provided detailed clarifications on the generative paradigm, added qualitative results for depth and pose, introduced a "generation-in-grid" strategy to improve NVS consistency, and provided attention map visualizations to explain the understanding mechanism.

**Reviewer Concerns:**

## Addressed:

- The authors successfully differentiated Omni-View from prior works by highlighting the "generative" vs. "reconstructive" objective difference.

- Comprehensive training details (GPU hours, dataset composition) and architectural specs were provided.

- Visualizations of depth estimation and outdoor generalization were added

- Token Activation Maps were provided to visualize how generation tasks broaden the model's attention

## Outstanding:

- Reviewer GSip noted limitations in long-sequence generation. The authors improved this with a "grid" strategy but acknowledged that perfect consistency remains future work.

- There remains a performance gap in grounding tasks compared to methods using explicit 3D inputs, which the authors acknowledged and analyzed.

**Reviewer Scores:**

- Reviewer Meww 6 ==> 8 (increase).  Very positive

- Reviewer Lkur:  8 ==>  8 (unchange). Very positive

- Reviewer U2JG: 4 ==> 6 (increase).  The rebuttal works and turn the scores from 4 to 6.

- Reviewer GSip: 6) ==>  6 (unchange). Positive.

The rebuttal is comprehensive.  All reviewers are supportive to accept this paper after the rebuttal.

---

### Decision · Program_Chairs · 2026-01-26

Accept (Poster)